# A Semi-Automatic Semantic-Model-Based Comparison Workflow for Archaeological Features on Roman Ceramics

Florian Thiery [1,*], Jonas Veller [2], Laura Raddatz [2], Louise Rokohl [3], Frank Boochs [2] and Allard W. Mees [1]

1   Leibniz-Zentrum für Archäologie (LEIZA), Department of Scientific IT and Research Software Engineering, 55116 Mainz, Germany; allard.mees@leiza.de
2   i3mainz—Institute for Spatial Information and Surveying Technology, School of Technology, Hochschule Mainz University of Applied Sciences, 55128 Mainz, Germany; jonas.veller@hs-mainz.de (J.V.); laura.raddatz@hs-mainz.de (L.R.); frank.boochs@hs-mainz.de (F.B.)
3   Leibniz-Zentrum für Archäologie (LEIZA), 55116 Mainz, Germany; louise.rokohl@leiza.de
*   Correspondence: florian.thiery@leiza.de

**Abstract:** In this paper, we introduce applications of Artificial Intelligence techniques, such as Decision Trees and Semantic Reasoning, for semi-automatic and semantic-model-based decision-making for archaeological feature comparisons. This paper uses the example of Roman African Red Slip Ware (ARS) and the collection of ARS at the LEIZA archaeological research institute. The main challenge is to create a Digital Twin of the ARS objects and artefacts using geometric capturing and semantic modelling of archaeological information. Moreover, the individualisation and comparison of features (appliqués), along with their visualisation, extraction, and rectification, results in a strategy and application for comparison of these features using both geometrical and archaeological aspects with a comprehensible rule set. This method of a semi-automatic semantic model-based comparison workflow for archaeological features on Roman ceramics is showcased, discussed, and concluded in three use cases: woman and boy, human–horse hybrid, and bears with local twists and shifts.

**Keywords:** linked open data; Wikidata; archaeology; close-to-original 3D digital objects; image matching; semantic modelling; rectification; interdisciplinary research; ceramics

## 1. Introduction

We are currently within the digital transformation era. Especially in the digital humanities and digital archaeology fields, together known as archeoinformatics, researchers and research software engineers (RSE) are developing and using new digital technologies to enhance traditional analogue science methods. In the so-called first wave of digitisation, machine-readable data have already been recorded and processed via web and cloud computing technologies. In the second wave of digitisation, we are required to analyse, enhance, and use these research data in multiple ways as machine-interpretable data using Artificial Intelligence (AI) technologies such as Machine Learning (ML), Deep Learning (DL), and semantic reasoning (SR) [1], representing an active part of archaeological development from the Analogue Era to Knowledge Graph Computing; which we might call Archaeology 4.0 [2].

Within Roman pottery research, research on African Red Slip Ware (ARS) has been massively hindered by the lack of a methodological framework to address the issue of feature comparisons. The lack of digitised objects and annotated machine-readable resources has obstructed the usage of AI and ML methods. Due to the lack of such a methodological framework which would enable us to address the complex issues of pottery workshop relations and chronologies, most of the hitherto published research is limited to iconological and art historical aspects of the imaging world depicted on these vessels.

The African Red Slip Ware digital project (ARS3D) addressed this issue by digitising and semantically annotating the ARS collection of the Leibniz-Zentrum für Archäologie (LEIZA),

formerly Römisch-Germanisches Zentralmuseum (RGZM). Within this project, ARS objects were digitised in 3D, semantically annotated based on the CIDOC CRM, and published online as FAIR data. With this new data basis, AI and ML methods are possible, which allow for verifiable statements of feature identifications, a precondition for Open Science.

This paper is a case study of the generic development and application of digital methods in archaeology based on the analysis of ARS. AI technologies in archaeology are rare (e.g., in numismatics [3–5]) and methods that can address the issues in ARS studies do not exist. ARS is not a research subject of itself in this study. Therefore, the viewpoint bias focuses on FAIRification and AI in archaeology, which enables us to provide answers to the complex chain of archaeological research questions involved. The digitisation and semantic annotation of ARS, along with the usage of AI, especially decision trees and semantic reasoning, provide a basis for developing new digital research methods within the archaeological community, as addressed in Section 2.

### 1.1. From an Analogue World to the Knowledge Era

Digital archaeology is part of the digital transformation from analogue to digital humanities science using Knowledge Graphs and Machine Learning methods. This consecutive and dynamically interacting process of era paradigms must consider the contributions and challenges of each paradigm within its social and research context. Each era creates relevant knowledge within the research community in which it is embedded.

These era paradigms (Figure 1) include the Analogue Era, which dates back to the first written records of human society; the Digital Era, which started in the mid-19th century AD with the publication of the first algorithms, e.g., Ada Lovelace [6]; the development of modern Turing-complete computers by Konrad Zuse [7] in the mid-20th century AD, and the development of the World Wide Web (WWW) as a "universal linked information system" [8] by Sir Tim Berners-Lee at CERN around 1989. The WWW serves as a base for the Semantic Era, initiated by creating the concept of Linked Data in 2006, a concept described as "making links so that a person or machine can explore the web of data" [9]. These techniques and inventions have led us to the current Knowledge Era, which deals with Artificial Intelligence techniques for the creation of new knowledge.

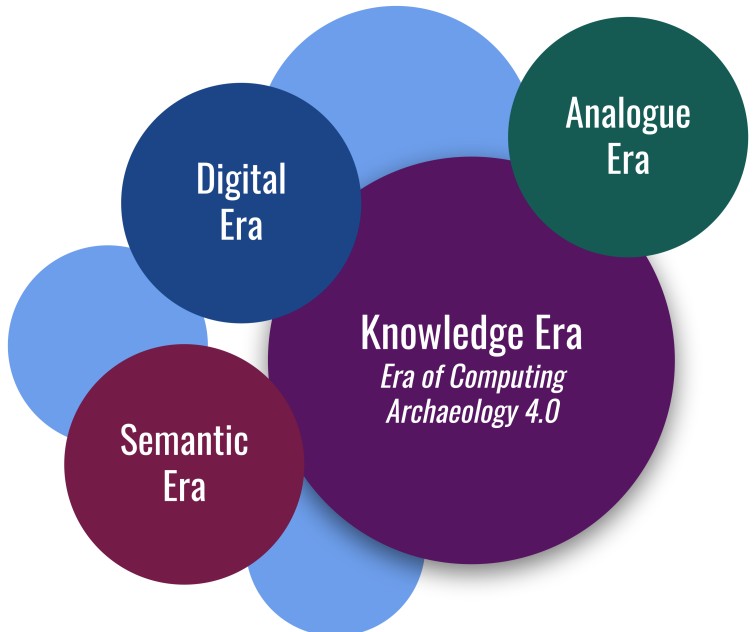

**Figure 1.** Schematic Model of paradigms related to development from an analogue world to the knowledge era (Florian Thiery, https://creativecommons.org/licenses/by/4.0/ (accessed on 9 March 2023), CC BY 4.0, via https://commons.wikimedia.org/wiki/File:Paradigms_Knowledge_Era.png (accessed on 9 March 2023), Wikimedia Commons).

Especially in archaeology, these era paradigms are characterised by research data stored in periodicals and books (Analogue Era), applying digitisation processes where research data are digitised and published online (Digital Era), applying semantic modelling and the publication of Linked Open Data [9] (LOD) in cooperation with community hubs, e.g., Wikidata, especially the WikiProject Archaeology [10] (Semantic Era), and applying AI technologies and semantic reasoning to generate knowledge (Knowledge Era).

The Knowledge Era can be called the Era of Computing, which comprises three paradigms: (I) programs, scripts, and procedures; (II) AI techniques such as Machine Learning; and (III) Knowledge Graphs, which use the technologies of Eras I and II (Figure 2). Within the era of scripts and procedures (Era I), developers hand-code step-by-step procedures by taking raw data and coming up with answers. Specifically, if we ask a program why we produced a specific answer, we can trace back the decision to a set of particular rules, which makes the system easy to explain. In Era II, developers do not write explicit *if–else rules* for each data byte in the input. Instead, they provide a training set of answers and the machine *learns* a set of complex rules. These rules are typically stored as a set of weights that are applied to input data as they move through a network. However, one cannot ask the system to explain why it made various decisions. The system only informs us about the numerical weights used to determine the answer. The Knowledge Graph Era (Era III) combines the first two eras of computing to produce systems that learn from complex data and explain their decisions. Here, developers continue to use ML techniques to harvest raw data and look for patterns in these data to convert them into new entries in a Knowledge Graph along with confidence weights. These data are then checked for consistency and quality by graph algorithms. What emerges from the graph is new knowledge (e.g., by applying semantic reasoning), answers, and explanations regarding why specific decisions are made. As a result, the Knowledge Graph becomes a repository of semantically precise nodes and edges with confidence weights retained from the Machine Learning processes. Nevertheless, these knowledge enrichment processes are imperfect, and can easily add false assertions if new knowledge is not curated by subject matter experts [11].

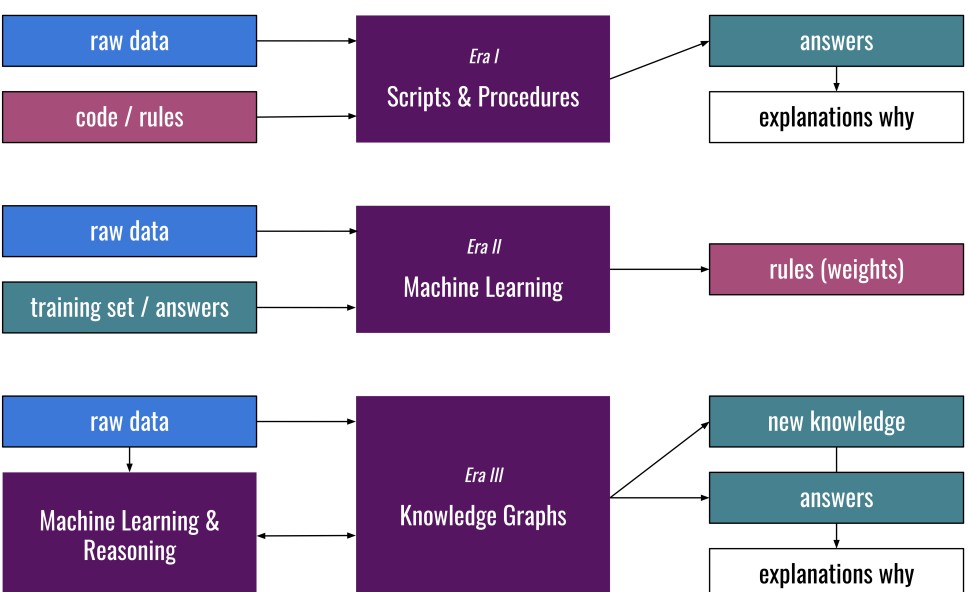

**Figure 2.** The *Era of Computing*, based on Dan McCreary [11] (Florian Thiery, https://creativecommons.org/licenses/by/4.0/ (accessed on 9 March 2023), CC BY 4.0, via https://commons.wikimedia.org/wiki/File:Three_Eras_of_Computing.png (accessed on 9 March 2023), Wikimedia Commons).

However, as the third era of computing and the AI approaches, such as ML or reasoning, we need a semantically modelled backbone using thesauri and ontologies

(Figure 3) [12]. Such semantically structured domain-specific data may help to create a context and serve as translation tools between languages, e.g., as applied in text mining and NLP applications in archaeology by Alex Brandsen [13–15].

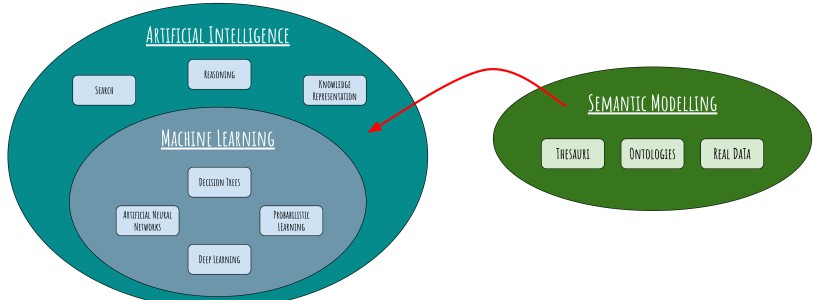

**Figure 3.** Semantic Modelling and Artificial Intelligence (Florian Thiery, https://creativecommons.org/licenses/by/4.0/ (accessed on 9 March 2023), CC BY 4.0, via https://commons.wikimedia.org/wiki/File:AIplusSemanticModelling.png (accessed on 9 March 2023), Wikimedia Commons).

### 1.2. Archaeology on the Way to the Knowledge Era

Combining semantics and AI technologies leads to the third *Era of Computing* [11], applying, for example, (semi-)automatic comparisons of pot forms, hallmarks, stamps, potter dies, and appliqués on Terra Sigillata using AI technologies such as semantic reasoning combined with semantic modelling in Knowledge Graphs.

There are different methods based on archaeological considerations and geometric comparisons that lead to, e.g., hypotheses related to the questions of whether or not figure types stem from moulds, or of whether there are misprinted versions of the same figure stamp or impressions of broken patrices. These methods can be entirely analogue (Analogue Era), such as visual inspection and creating rubbings or drawings of forms, inscriptions, and figure types; cf. Figure 4.

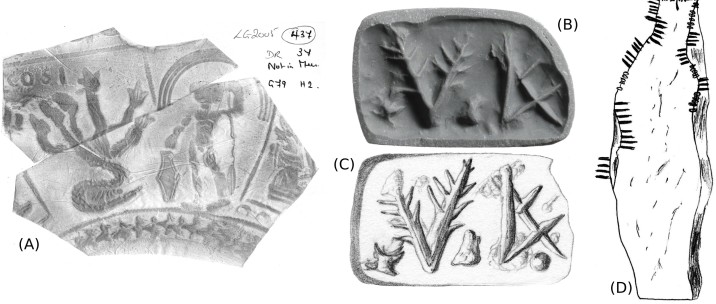

**Figure 4.** (**A**) Rubbing of a decorated Samian (Terra Sigillata) vessel made in La Graufesenque, 100-120 AD, decorated by the potter L. Cosius (Geoffrey Dannell/Allard W. Mees, https://creativecommons.org/licenses/by/4.0/ (accessed on 9 March 2023), CC BY 4.0, via https://commons.wikimedia.org/wiki/File:Rubbing_Samian_Potter_L._Cosius.png (accessed on 9 March 2023), Wikimedia Commons). (**B**) Seal with hole, Minoan (MET, 26.31.98), CMS No. XII 087a [16] (p. 146), Q61293075 (http://www.wikidata.org/entity/Q61293075, accessed on 9 March 2023) (Metropolitan Museum of Art, CC0, via https://commons.wikimedia.org/wiki/File:Seal_with_hole_MET_DP110967.jpg (accessed on 9 March 2023), Wikimedia Commons). (**C**) Drawing of CMS No. XII 087a [16] (p. 146), with CHIC 029 (category V, Végétaux [17] (pp. 15–17)), entity 1160613 (Siegel CMS XII 087a) in iDAI.objects (https://arachne.dainst.org/entity/1160613 accessed on 9 March 2023) (University of Heidelberg/CMS, CC BY, via https://arachne.dainst.org/entity/6900574 (accessed on 9 March 2023), iDAI.objects). (**D**) Ogham stone drawing of CIIC 81 with inscription *C[A]SSITT[A]S MAQI MUCOI CALLITI* (Florian Thiery, https://creativecommons.org/licenses/by/4.0/ (accessed on 9 March 2023), CC BY 4.0, via https://commons.wikimedia.org/wiki/File:CIIC_81_by_Macalister_(1945).png (accessed on 9 March 2023), Wikimedia Commons).

These catalogues, which are mostly books by Hayes [18], Armstrong [19], zu Löwenstein [20], Atlante [21], Macalister [22], McManus [23], and Ziegler [24], are the basis for digital publication and research (Digital Era), such as distribution maps of Samian Ware by specific potters and stamps, also called dies (here: *bowls by Aquitanus with die 2a*), or Ogham inscriptions (here: *formula word MUCOI, engl. belongs to tribe*); cf. Figure 5. The huge challenge in the case of digital transformation is the problem that while these data are well structured and processed, they are not made available in a semantically structured, machine-readable (Semantic Era), and *FAIR* (Findable, Accessible, Interoperable, Reusable) way [25,26]. Here, the domains of numismatics, ceramics, Ogham inscriptions, and seals are leading the way in the publishing and connecting of research data as Linked Open Data [27,28] using community hubs, e.g., *Nomisma* (http://nomisma.org, accessed on 9 March 2023), *Kerameikos* (http://kerameikos.org, accessed on 9 March 2023), *archaeology.link* (http://archaeology.link accessed on 9 March 2023), and *Wikidata* (https://www.wikidata.org, accessed on 9 March 2023). In addition, this includes domain-specific research collections and Wikidata projects as part of the Collection Research Network [29] (p. 2), such as *Roman Open Data* (https://romanopendata.eu, accessed on 09 March 2022), *Linked Open Samian Ware* [30–33], *Linked Ogham Data* [34–40] (pp. 119–127), the *Wiki-Project Prähistorische Keramik* [41,42], and *Linked Aegean Seals* [43] within the Corpus of the Minoan and Mycenaean Seals (CMS) project (http://cmsheidelberg.uni-hd.de, accessed on 9 March 2023) using the iDAI.world (https://idai.world, accessed on 9 March 2023) infrastructure, i.e., iDAI.objects, iDAI.chronontology, and iDAI.gazetteer. This does not enable these data to be used in quantitative classification methods in archaeology and further AI approaches such as ML, DL, neuronal networks, or semantic reasoning (Knowledge Era).

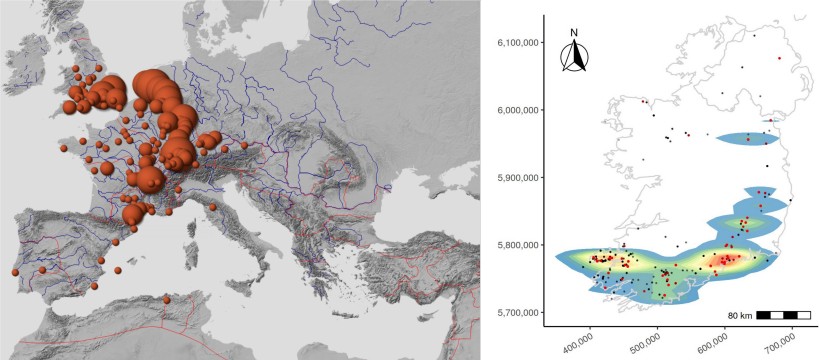

**Figure 5.** (**Left**) Samian Ware distribution map from the Samian research database (2nd of March 2023) of bowls in Gaulish-Germanic-Reatian tradition, created in kiln site La Graufesenque by potter Aquitanus (Samian Research Community, https://creativecommons.org/licenses/by/4.0/ (accessed on 9 March 2023), CC BY 4.0, via https://commons.wikimedia.org/wiki/File:Samian_Ware_La_Graufesenque_Aquitanus.png (accessed on 9 March 2023), Wikimedia Commons); (**right**) density map of the word *MUCOI* (engl. "tribe" or "sept") on Ogham stones (Sophie C. Schmidt, https://creativecommons.org/licenses/by/4.0/ (accessed on 9 March 2023), CC BY 4.0, via https://commons.wikimedia.org/wiki/File:Ogham_Densitymap_German.jpg (accessed on 9 March 2023), Wikimedia Commons).

### 1.3. Archaeology and Artificial Intelligence

Within archaeology, exploration of the usefulness of AI techniques was first begun about ten years ago. ML/DL algorithms and neural networks were introduced in sessions dedicated to AI at archaeological conferences such as the Computer Applications and Quantitative Methods in Archaeology (CAA) conference, currently S13 at CAA 2022 [44] or S21 at CAA 2019 [45]. AI methods and techniques are currently applied in archaeology for, e.g., the discovery of archaeological sites, the recognition and reassembly of archaeological pottery, the extraction of text and named-entity recognition, and iconography. Overviews of applied AI techniques in archaeology are provided in Gualandi et al. [46] and Bickler [47].

Examples that illustrate AI technologies include applying or enabling comparisons of parts of archaeological artefacts, which can be applied in the ceramic research domain and in numismatics.

Ponciano et al. [48] demonstrated automatic detection methods of objects in 3D point clouds based on semantic-guided processes and connected semantic concepts as a basis for the recording and computer-based modelling of cultural heritage (CH) objects, and thereby as a basis for point cloud-driven comparisons in a 3D space [49].

The ArchAIDE project (http://www.archaide.eu, accessed on 9 March 2023) implemented an AI-based application to recognise archaeological pottery in a 3D space. The technique relies on the shape of a potsherd, and is based on decorative features [46]. This project empowers 3D models of archaeological artefacts to compare and cluster them using the pot form, which is a common archaeological standard method.

Poux et al. [50] developed a process that integrates semantics into the segmentation of point cloud data and the classification of detected objects. This allowed them to enrich spatial data with different types of knowledge and to support processing with semantic content ranging from acquisition characteristics to archaeological facts, with the benefit of clearly improving the classification results.

Many ML/DL methods on 2D raster images are applied in numismatics, serving as a basis for comparing and clustering coins and their depictions. The Big Data Lab at Goethe University Frankfurt am Main has applied and implemented a variety of recognition and implicit comparison approaches in student theses such as Loyal [3], Krause [5], and "Combining machine learning methods of image and text recognition on ancient coin data" [4]. Gampe described the considerable challenges in the condition of the coins and the training data. However, the greatest problem with image recognition is the network's focus on the inscriptions on the coins instead of on the portrait. A survey of image-based coin recognition systems has been provided by Sasi and Sreekumar [51]. A blog post by Steinberg et al. [52] described how to build a coin recognition system from scratch.

An example of an AI approach based on semantic reasoning combined with Linked Open Data (LOD) is the Academic Meta Tool (AMT) [53]. AMT is a methodological proposal for modelling vagueness in graphs using a specific ontology, and can help to compare potter dies. The JavaScript library can perform reasoning with several logics. For example, the AMT approach has been used to model a South Gaulish Terra Sigillata pottery network with the AMT concepts Potter and Die. With axioms, one can determine several pieces of information. For example, if a potter created a die, it can be automatically rule-based from a surmoulaged die that the potter eventually worked in the same workshop [53] (Section 5.2).

While details such as inscriptions on coins are difficult to observe in 2D raster images, this information can become visible in high-resolution 3D models, as the third component, the depth/height of the objects in the 3D models, is known. Optical 3D close-range scanners provide high accuracy for detailed analysis of objects. For example, Mara [54] used high-resolution 3D models to improve the readability of ancient stone inscriptions on cuneiform tablets. In addition, filter and pattern recognition methods have been developed based on 3D data, and enable the automatic extraction of cuneiform characters. Writing and other details on objects, such as damage (cracks, spalling) or ornamentation, can be recognised and analysed in high-precision 3D models.

Precise digital 3D models are an essential factor in industry, and are mainly used for quality control [55]. For example, 3D models of produced components are created and a surface comparison with the corresponding constructed design model (CAD model) is carried out to check how high the deviations from the CAD model are and whether they are within the tolerance range. After a surface comparison, minor deviations can be displayed in colour. The colour scale can be adjusted as desired, allowing even the slightest deviations to be made visible [56,57]. This comparison method can be transferred to archaeology, except that there are usually no nominal dimensions (no CAD model) for archaeological objects. Instead, two similar objects or parts of objects can be compared with each other or an object can be compared in different states, e.g., before and after

conservation [58]. Through 3D digitisation, a data basis is created that offers the potential for many analysis possibilities.

### 1.4. The ARS3D Project

In order to illustrate the possibilities of the developed comparison workflow, we use the example of African Red Slip Ware (ARS), particularly the collection at the Leibniz-Zentrum für Archäologie (LEIZA), formerly known as the Römisch-Germanisches Zentralmuseum (RGZM). ARS is a type of Terra Sigillata, or fine ancient Roman pottery. Its striking decoration characterises itself through relief decorations (appliqués) depicting mythological, Old Testament and New Testament scenes, circus, arena, and hunting depictions, and fish and plant images (Section 2.2) [18–20]. This exceptional pottery type offers unique knowledge potential for cultural history questions, especially for the creation of Digital Twins (Section 2.3), i.e., digital capturing, semantic modelling, and AI-based comparison processes. We define a Digital Twin as a virtual representation that is the digital counterpart of a physical object. The African Red Slip Ware digital (ARS3D) project (https://ars3d.rgzm.de, accessed on 9 March 2023) was funded by the German Federal Ministry of Education and Research (BMBF) in 2018–2021. ARS3D aims to provide the research community with innovative and sustainable access to an essentially digitally unpublished object type, both to pose new research questions and to set documentation standards. As of March 2022, the ARS3D use case contains a corpus of 336 ARS objects from the LEIZA collection. During the project, Digital Twins of these objects were created by digitising the objects in 3D and semantically modelling their features. Based on these data, digital processes were developed that enable various scientific analyses [59]. Thus, for example, comparisons between appliqués can be carried out, which can provide a basis for statements about the same manufacturing processes of ARS.

This paper addresses issues related to the following question: *To what extent can semi-automatic semantic model-based geometric comparison methods be combined in interaction with analysis of archaeological features on Roman ceramics? Such combination would allow the archaeological research community to obtain new conclusions about ancient manufacturing techniques and relations between ancient workshops.*

First of all, such comparisons require precise object models from which archaeologically and geometrically oriented information can be derived. Positive and negative representations of the objects can be used for this purpose. There are various techniques for this in different object genres, such as ceramics or coins, including appliqué and mould, embossing and stamps, or matrix and patrix. One of the archaeological challenges in semantically describing and annotating objects and appliqués is the relatively small corpus of partially fragmented objects, which causes problems in defining fully comprehensive comparison rules. The problem is that no objects in this corpus are known to fit together geometrically, e.g., appliqués with matching moulds or stamps with matching manufacturing tools. However, in other collections, parallels are thought to be present; therefore, establishing a base for future comparisons is required. We take the following approach: as a first step, we compare appliqués with the hypothesis that geometrically identical objects come from the same manufacturing place, thereby enabling conclusions to be drawn about the geographical origin of the workshop involved, its dating, and the manufacturing process. There has been only an overseeable amount of experimental archaeology for Gaulish Red Slip Ware on the subject of figure stamp copying by surmoulage and the clay shrinkage and eventual deformations involved [60]. However, its observations are helpful in establishing criteria regarding whether a specific figure stamp (patrix) was, e.g., used in comparable pottery traditions, such as African Red Slip Ware. One of the core questions in research on these potteries is whether a figure stamp observed in the context of another pottery is in fact a surmoulage or not [61]. The physical movement of a patrix to another pottery points either towards a "wandering potter" [62,63] (pp. 346–347), or alternatively to the sale or rent of the figure stamps (patrices) themselves, which could have belonged to an inventory of rented potteries, as indicated in contemporaneous Roman Egyptian potter contracts

preserved on papyri [63] (p. 258, Abb. 175). The ownership changes of figure stamps can even be detected within the same pottery when figure stamps appear in different pottery consortia within the same pottery [63] (p. 305, Abb. 210). Whether a figure stamp observed at different workshops was copied by surmoulage or handed over in a damaged version is of pivotal importance in establishing chronology in potteries; if a broken version of a figure stamp appears in another pottery, it is clear that the latter is chronologically younger and the still-complete version is older [63] (Abb. 25). With surmoulages this is less clear, as the original figure stamp from which the copy was taken may have remained in use; therefore, in this case no clear chronology can be established [63] (Abb. 23). In addition, the fundamental challenge is to consider the possible technical modifications in the objects after they have been produced (taking them out of the mould, applying them, using the plates, vases, ageing, etc.), which are unknown in their impact and difficult to model. This modification results in possible geometrical distortions, making a comparison with another object difficult, as no complete identity exists. Therefore, a comparison process (appliqué–appliqué, appliqué–form) must be tolerant of differences. Furthermore, appliqués are potentially geometrically deformed, as they are applied to carrier objects (vases, bowls, plates, etc.) in geometries that cannot be unwound without deformations, and are partly discontinuous. These geometric aspects prevent a fully automatic comparison process. This is why an interactive, semi-automatic, and semantic model-based comparison workflow for ARS features has been developed, including archaeological domain-specific aspects, as an attendant aid for modelling comparison interpretations (Section 2.4). To make the data reusable and provide comprehensible FAIR data, this comparison workflow has to cover certain aspects: (i) geometric steps must be structured and modelled; (ii) archaeological aspects must be described and structured according to standards; and (iii) the comparison process must bring all contents together in a semantic framework (Section 2.5).

This paper describes the used materials and methods (Section 2) as well as three comparison results (Section 3), and discusses the methods and results (Section 4). To conclude, we look at the remaining challenges (Section 5) for this approach to a semi-automatic semantic model-based comparison workflow for archaeological features on Roman ceramics. The online links provided in the text were last accessed on 9 March 2023.

## 2. Materials and Methods

This section describes the used materials (Section 2.1) from an archaeological collection at the LEIZA; in Section 2.2, we describe the general concept of our multi-stage approach (Figure 6), followed by the methods used to create a Digital Twin (Section 2.4), the individualisation and comparison of objects (Section 2.5), and the semantic modelling approach (Section 2.6).

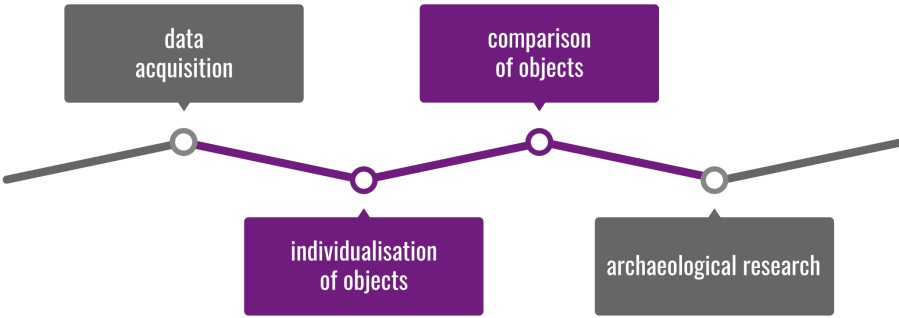

**Figure 6.** Scheme of the multi-stage approach for analysing archaeological materials (Florian Thiery, https://creativecommons.org/licenses/by/4.0/ (accessed on 9 March 2023), CC BY 4.0).

### 2.1. African Red Slip Ware (ARS)

A special challenge in the archaeological documentation of objects is represented by late Roman decorated Red Slip Ware vessels. The representations attached to these

vessels using moulded relief appliqués cannot be adequately recorded with traditional 2D documentation methods due to different vessel curvatures. In addition, there is no standardised means of addressing the image representations in the relevant literature. As there are only a relatively small number of known decorated African Red Slip vessels (here, ARS, sometimes abbreviated as ARSW in the literature) (cf. Section 2.2), elaborate 3D documentation is possible, and a semantic annotation model can cover a significant percentage of the extant material.

This tableware is characterised by its red slip body clay, with the slip usually being one shade darker than the body [18]. Various Red Slip Wares can be divided into several types from different production areas, as applied by Lizanne Mollema [64] (pp. 11–17): (i) African Red Slip Ware (ARSW), known as Late Roman A and B pottery, Terra Sigillata Chiara, or Terra Sigillata Africana; (ii) Cypriot (CRSW), known as Late Roman D Ware [18] (p. 371); (iii) Egyptian (ERSW), known as "imitation" ARSW, where production only began when the former had already become popular in the region; (iv) Phocean (PRSW), known as Late Roman C [65]; and (v) Sagalassos (SRSW), a relatively new addition to Red Slip Ware [66].

The material subgroup being studied here, Sigillata C, is a fine ceramic genre of ARS produced in the mid-3rd century AD in present-day Central Tunisia, with the quality of the ware and the chronological order serving as diagnostic criteria. This ceramic genre can be divided into five subgroups (C1–C5). Subgroups C3 and C4 can be assigned to Late Antiquity. The repertoire of forms includes bowls, plates, and platters. Jugs and pitchers, on the other hand, are rather rare. The vessels are either undecorated or have decorations such as appliqués, relief, stamps, or chatter decoration. Sigillata C is a dense, fine-grained, and hard-fired clay with a colour varying from orange to red-orange or from red to brown-red. The engobe covers the entire vessel, and has an orange to red-orange colour [20] (pp. 398–399).

Frederick O. Waagé and Nino Lamboglia have proposed two classifications of ARS fabrics. On the one hand, Waagé distinguishes two types: Late Roman A (LRA, fine-grained smooth-surfaced wares typical in the third and fifth century) and Late Roman B (LRB, standard coarse-grained varieties); on the other, Lamboglia distinguished four types: t.s. Chiara A-D, with TSC-A, TSC-C, and TSC-D being African and TSC-B being Gaulish. These opposing fabrics indicate the main developments in ARSW, and are related by Hayes as follows: (i) LRB: early phase = TSC-A: late first–third century; (ii) LRA = TSC-C: third–fifth century; and (iii) LRB: middle and late phases = TSC-D: fourth–seventh century [18] (pp. 287–288). After Bes [67] (pp. 22–24), ARSW should be seen as an umbrella term under which a general fabric classification exists. Several production locations have been investigated archaeologically, and the ceramics have been characterised through archaeometric research, e.g., [68].

ARSW was mainly produced in large potteries in modern Northern and Central Tunisia (in the Roman provinces Africa Proconsularis and Byzacena) from the 1st to the 7th century [69] (pp. 121–122); cf. Figure 7. Sites of potential ARSW production have been investigated in archaeology, including several characterised by archaeometric research, e.g., [68] (p. 151). This research focuses on regional production models enhanced by the diachronic and geographic development of pottery workshops [67] (pp. 22–23, Figure 7).

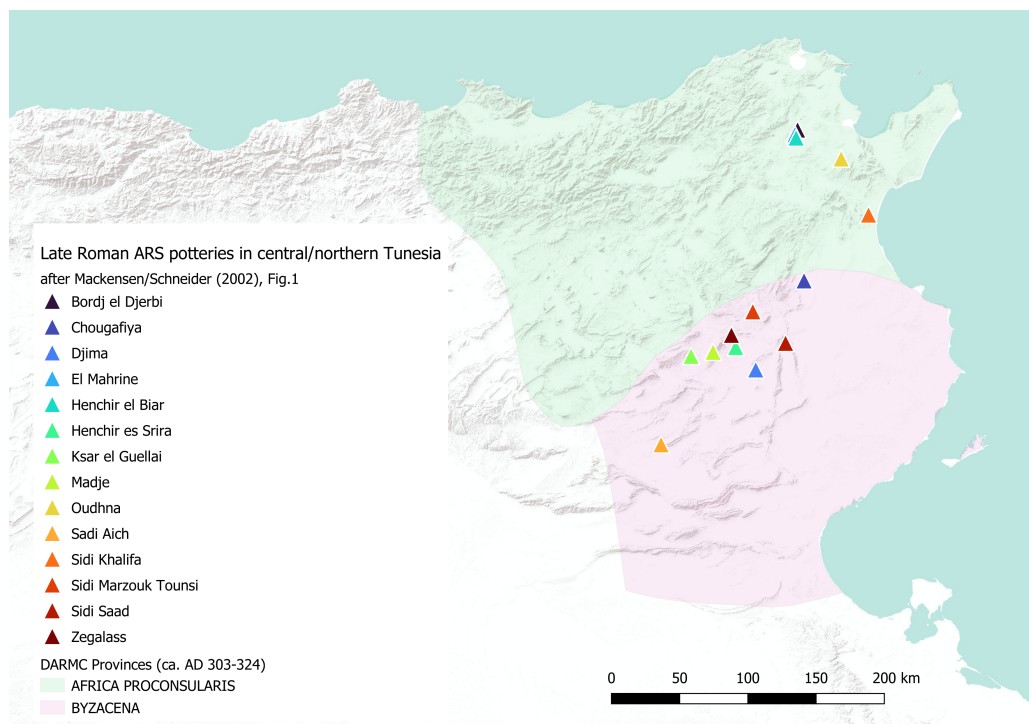

**Figure 7.** Late Roman ARS potteries in central and northern Tunisia, scale 1:3000000, EPSG:3859, bounding-box: 408172.541, 3881225.581, 1299172.542, 4511225.581, created with QGIS, base map ESRI.WorldTerrain by USGS, Esri, TANA, DeLorme, and NPS. The potteries have been georeferenced using Mackensen and Schneider, Journal of Roman Archaeology, Vol. 15, 2002, [69] (Figure 1) as a basemap. Province data by https://arcg.is/14yeuS (accessed on 9 March 2023), the Digital Atlas of Roman and Medieval Civilizations (DARMC) Roman World Data, layer: Roman Provinces (ca. AD 303-324), based on the Barrington Atlas [70] with contributions from Matthew Polk. The geodata is published by F. Thiery under multiple open licenses [71] (Florian Thiery, https://creativecommons.org/licenses/by/4.0/ (accessed on 9 March 2023), CC BY 4.0, via https://commons.wikimedia.org/wiki/File:Late_Roman_ARS_potteries_Tunesia.png (accessed on 9 March 2023), Wikimedia Commons).

The appliqué- and relief-decorated vessels of Sigillata C3/C4 were exported throughout the Mediterranean region. Apart from many finds in North Africa, the geographical distribution of finds is recognizable throughout the Mediterranean, with a focus on the east coast of Spain, Portugal, Sicily, and Italy, the south coast of France and behind the Pyrenees, as well as isolated finds from Sardinia and Corsica; see Figure 8 [18] (pp. 453ff.), [64,67] (Appendix A), [71,72]. However, such distribution maps are always dependent on the past and current states of archaeological research in different areas.

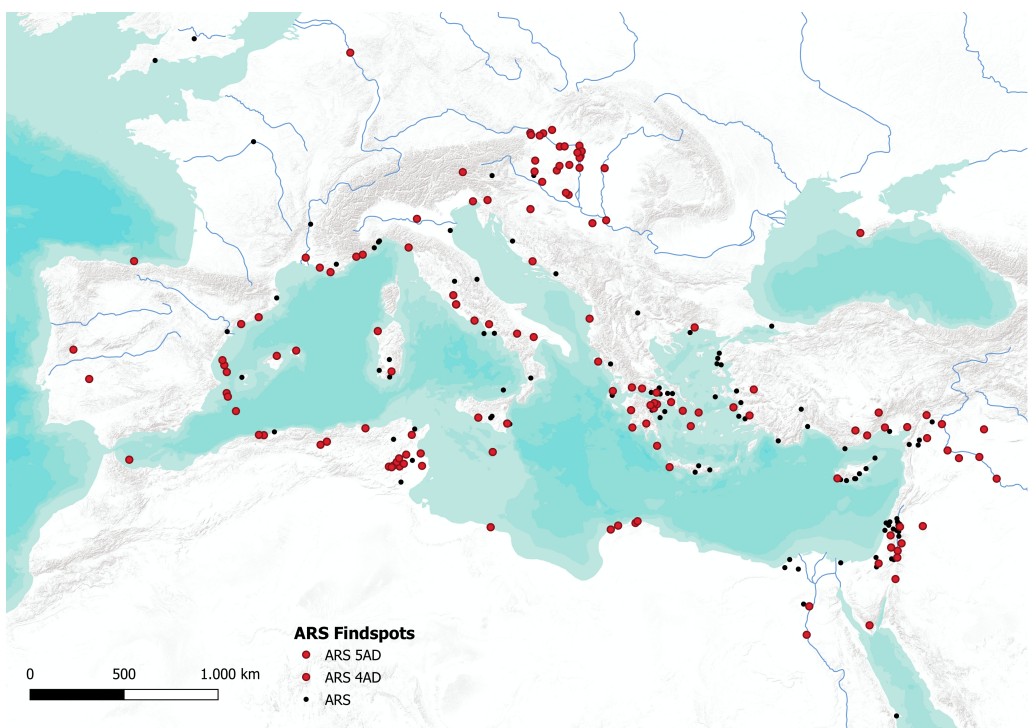

**Figure 8.** Distribution of ARS, scale 1:20000000, EPSG:3859, bounding-box: -1227354.449, 2644882.041, 4712645.557, 6844882.045, created with QGIS, rivers by geojson.xyz and Natural Earth Data (Public Domain), base map ESRI. WorldTerrain by USGS, Esri, TANA, DeLorme, and NPS. ARS data are based on Lizanne Mollema's bachelor thesis 2018 at Leiden University [64] (Appendix 1) (Lizanne Mollema, CC BY) and have been transformed into GeoJSON and WGS84 [72]. The geodata were published by F. Thiery under multiple open licenses [71] (Florian Thiery, https://creativecommons.org/licenses/by/4.0/ (accessed on 9 March 2023), CC BY 4.0, via https://commons.wikimedia.org/wiki/File:Distribution_of_ARS.png (accessed on 9 March 2023), Wikimedia Commons).

Absolute and relative chronological classifications of Late Antique appliqué- and relief-decorated Sigillata ARS C3/C4 vessel forms have only approximately been established [73] (pp. 16–17, Abb. 4). Various appliqué- and relief-decorated vessel forms had different durations [20] (pp. 400–401). Due to inscriptional and iconographical evidence on a few vessels themselves referring to the terms of specific Roman consuls, C3/C4 vessels can be dated to between the 3rd and 5th centuries AD [74] (p. 117). External features, such as the pot form, decoration technique, and sherd quality, as well as motifs, may serve as dating criteria for the primary production stages. Moreover, further criteria for chronological classification include iconographic clues and the choice of subject, e.g., hairstyles, jewellery, clothing, and the recurrence of certain themes. Within the Late Antique Sigillata C3/C4, two different techniques of relief decoration can be identified, namely, appliqué decoration and relief decoration formed entirely from models, for example, in [18] (pp. 211–214). The appliqué decoration is a characteristic feature of the Sigillata C1–C3 vessels. In this process, a clay mass was moulded into a matrix (negative mould), then the appliqués, which had shrunk by 8–10%, were removed from the mould in a leather-hard state, applied to the vessels with the help of fine clay slip, and finally engaged before firing. ARS seems to have been fired in kilns under well-controlled conditions [20] (pp. 410–412). Trails by M. Farnsworth indicate firing temperatures of around 1000 °C or rather below this [18] (p. 295). The process described was not without risk, as the appliqués could flake away from their base due to a varying degree of drying and water content.

Relief-decorated forms produced entirely from moulds are typical of, for example, the Hayes 56 Sigillata C4 plates, where the mould usually reveals the vessel's shape [75] (p. 115), [76] (Figures 4 and 5). Here, the rim may have been already decorated in the matrix (typical for C4 ware). A frequent variation was that the rim was left smooth and then appliquéd after the clay plate had been moulded (typical for C3 ware). These two types of decorated vessels

(mould-made appliqués applied to a vessel and vessels made in decorated moulds) may have been preceded by a moulding chain consisting of several working steps. Appliqués could have been made from moulds and impressed in series in new moulds from which, potentially due to shrinkage, new somewhat smaller appliques may have been made, thereby establishing a possible chain of several moulding generations (an original mould from which a workshop mould (patrix or matrix) was moulded, the workshop mould, and the final product). Figure 9 (left) shows two appliqués applied on vessel *bowl with erotes on board* (O.41260): a boat with fishing erotes *Mariner Thiasos, Fischende Eroten J/BT V* [20] (p. 710) (Q117034588) and a *fish swimming right* described in Armstrong 5.15 [19] (pp. 104–105) (Q117034496). Figure 9 (right) shows a patrix impression in a mould in order to create the applique on O.41418 of a *boating ship with erotes that are looking to the left*, described in G/FT III [77] (pp. 9–11) (Q111372294).

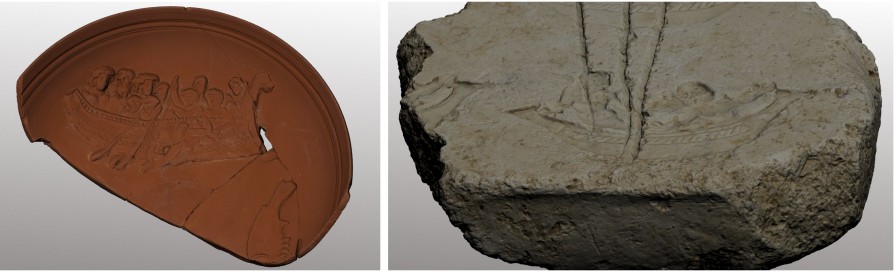

**Figure 9.** (**Left**) Appliqués applied to vessel O.41260; (**right**) patrix impression in a mould in order to create the applique on O.41418 (ARS3D project/i3mainz/RGZM, https://creativecommons.org/licenses/by-sa/4.0/ (accessed on 9 March 2023), CC BY-SA 4.0, via https://commons.wikimedia.org/wiki/File:ARS3D_O.41260_O.41418.png (accessed on 9 March 2023), Wikimedia Commons).

This highly specialised *chaîne opératoire* was based on a profound labour division typically known in all Red Slip Ware (as well as earlier Italian and Gaulish) workshops [63] (pp. 233–238; 299–316). Moreover, figure types and motifs may have been derived from, e.g., silver vessels or bronze caskets [76] (p. 10). Very early in ARS research it became clear that impressions for the production of C4 plate mould forms seem to have been derived from ivory diptychs [76] (p. 9), [74] (p. 118, Abb. 3–4), of which one even appeared to be historically datable [78] (p. 222), [74] (p. 117). How the potters involved obtained access to these precious materials remains, however, entirely unclear [73] (p. 15).

In addition to the two specialised production techniques described above, the figure type may have been stamped or individually carved with a stylus into the flat surface of a vessel that had already been moulded [20] (pp. 410–412), [73] (p. 15).

Therefore, the main and hitherto scarcely addressed archaeological question is whether or not almost identical appliqués were made from the same mould, or whether regular shrinkage between very similar appliques is discernable [79] (pp. 162–163, Abb. 22–23). In existing archaeological figure stamp catalogues, surmoulages are regularly indicated; however, it is seldom made clear on which categorisation the statement is based [80,81]. Only very precise documentation and observation rules may provide answers regarding whether they provide only a very close stylistical link (which does not automatically implicate a direct workshop link) or whether it demonstrates a workshop link whereby the same physical figure type was used on another form type or even in another workshop [53] (Chapter 5.2).

## 2.2. The RGZM/LEIZA ARS Collection and Its Provenance

There are four significant ARS collections known in Germany: (a) 148 objects known at the Romano-Germanic Museum in Cologne [82] (p. 413); (b) the State Collections of Antiquities in Munich [78,83], which comprise at least 240 pieces [78] (pp. 262–345); (c) the Archaeological Collection of the Bavarian State in Munich, which contains at least 200 pieces [84,85] (pp. 7–8); and (d) the LEIZA collection, which comprises at least 325 pieces. Collections a–d have only partially been published in analogue catalogues.

All these German collections compiled during the 20th century comprise large amounts of vessels from formerly private collections or that were acquired on the art market [78] (p. 18), [85] (p. 1), [82] (p. 413). Because a considerable number of complete vessels have been preserved, it is difficult to avoid the conclusion that they stem from looted cemeteries in Tunisia, as complete vessels are usually not found during regular excavations [78] (p. 19). This means that most vessels have already been de-contextualised in Tunisia, as any further information of other finds in the looted graves is entirely lacking [73] (pp. 22–23). However, despite the disputable provenance of these objects, their frequent completeness offers a unique opportunity to create accurate 3D representations, handle the issue of rectifying figure types on irregularly curved hemispheres, and make them available as Open Data in order to establish "Knowledge Equity" [86,87]. This is a core topic of the Wikimedia Foundation [88], which has set up, e.g., scholarship programs such as the Open Science Fellows Program [89–91], comprising projects [92] on Knowledge Equity and Open Data, such as P. Schrögel on the participation and diversity in Open Science [93], J. Bemme on European local history research [94], S.C. Schmidt on archaeological sources in Wikidata [95], and F. Thiery on Irish Ogham stones in the Wikimedia Universe [96]. Furthermore, Wikimedia offers community hubs and freely available services such as Wikidata, Wikipedia, or Wikimedia Commons, providing Open Data without any licensing or power barriers.

Although the United Nations *Convention on the Means of Prohibiting and Preventing the Illicit Import, Export and Transfer of Ownership of Cultural Property* was already established in 1970 [97], legalisation in Germany by the Kulturgüterrückgabegesetz (KultGüRückG) was implemented only in 2007 [98,99]. Virtually all of the materials in this study came to Germany longer than 30 years ago, which made §11(1) KultGüRückG applicable, according to which it is legally not compulsory to return cultural materials to their country of origin. This delay in legal action was heavily criticised by the public and by the RGZM [100]. This helped to raise public awareness concerning this issue, and a new research discipline, "Provenance Research", emerged [73] (p. 22, fn. 28), [101,102]. During the following years, data related to the emerging Open Data and FAIR approaches, definitions of *CARE* (Collective Benefit, Authority to Control, Responsibility, Ethics) [103,104] were established [26,105]. As a result, institutions such as the LEIZA nowadays do not purchase anything on the art market and are focused on producing analogue copies in their restoration and conservation departments, as well as 3D documentation in specialised documentation units. German national funding bodies (e.g., BMBF or Deutsche Forschungsgemeinschaft) are encouraging this development, and large funding programs such as ARS3D have been launched in order to create and publish open virtual research platforms. In this way, the countries of origin of the displaced objects can have access to these digital copies, for which purpose open licences are being used.

At the RGZM, the acquisition of ARS started in 1913, when a vessel [106] (p. 124) [18] (O.7046, Hayes form type 75) was bought from an Italian antiquarian. Roughly a third of the RGZM collection was, in the 1970s, obtained from a previously private collection that belonged to the Dutch archaeologist Jan Willem Salomonsen [107] (pp.17–21) [108] (p. 291), [109]. While "Unfortunately, there is only limited information on the survey campaigns of Salomonson available" [107] (p. 17), there are indications that part of the material from the Salomonsen collections was found at the ARS production site *Sidi Marzouk Tounsi* in Tunisia [78] (p. 41); cf. Figure 7. A considerable number of further purchases were made on the art market. These acquisitions themselves were usually documented and published [110] (pp. 656–657) and the provenance of these objects was usually unclear.

The ARS3D project has captured ARS objects from the LEIZA (former RGZM) collection (n = object count, c = in brackets, number of complete objects): (i) objects in total n = 325 (c = 52); (ii) object types: African Red Slip Ware n = 271 (c = 45), manufacturing objects n = 54 (c = 7), i.e., mould, stamp; (iii) shapes: bowl n = 157 (c = 18), dish n = 64 (c = 3), lamp n = 27 (c = 19), plate n = 7 (c = 1), jug n = 6 (c = 4); (iv) dating: Late Antiquity according to Hayes [18] (p. 13ff.) and zu Löwenstein [20] (pp. 400–401); (v) pot forms Hayes 53A [18] (pp. 78–82): n = 141 (c = 16), Hayes 56 [18] (pp. 83–91): n = 37 (c = 0), Hayes 54 [18] (pp. 82–83): n = 5 (c = 0), Atlante X A1a [21] (pp. 200–203): n = 15 (c = 12); and (vi): findspot

is unknown, expected Northern Africa, Tunisia. The nature of the LEIZA collection and its various archaeological objects and artefacts pose challenges for data capture as well as for data modelling, as explained in Sections 2.4.1 and 2.4.2.

### 2.3. General Approach and Methods

The data basis is provided by close-to-the-original 3D digitisation subject to various analyses and semantic modelling. To implement 3D analyses, geometry and semantic tools are required and are integrated into web applications. Furthermore, computational processes are necessary to extract appliqués from the entire 3D model and to rectify and compare them. These challenges lead to a multi-stage approach, at the end of which a web-based query tool is made available to study, analyse, and compare ARS objects with other ARS objects in a multi-layered way. To compare potentially similar objects, hypotheses can be formulated and evaluated based on logical rules. This approach involves several steps, with each providing necessary content and intermediate results (Figure 10).

| Step | | Result / Output |
|---|---|---|
| Data acquisition | Close to the original capture of an ARS object | textured 3D model |
| Individualisation of objects | 3D visual inspection and archaeological investigation | archaeological knowledge formation |
| | 3D visualisation and extraction of individual appliqués | 3D model of an appliqué |
| | Capture of archaeological features of individual appliqués | archaeological object data |
| | Geometrical correction of an applique for the influence of the carrier object | 2.5 D geometry of an applique incl. quality values |
| Comparison of objects | Geometrical comparison of selected pairs of appliqués | detailed, areal evaluation of the geometrical conformity |
| | Visualisation of the comparison result | basis for an archaeological assessment of the match |
| | archaeological assessment | supplement to the geometric assessment result |
| | rule-based assessment of a similarity hypothesis | enriched specialist data |
| Archaeological research | web-based query | provision of all contents for specialist queries |

**Figure 10.** Steps, results, and output of the multi-stage approach for African Red Slip Ware comparisons (Prof. Dr. Frank Boochs, https://creativecommons.org/licenses/by/4.0/ (accessed on 9 March 2023), CC BY 4.0).

### 2.4. The "Digital Twin"

We can define a Digital Twin as a virtual representation that is the digital counterpart of a physical object. In our case, Digital Twins can be seen as a synonym for digital models or shadows. A popular example of Digital Twins is 3D modelling to create digital companions for archaeological artefacts combined with the knowledge modelling of expert data. A Digital Twin is a copy of certain physical properties, including shape, texture, and meta- and paradata. The idea of the Digital Twin was further developed by M. Grieves and J. Vickers into a conceptual model of the Digital Twin, which consists of three main parts: (i) the physical products in real space; (ii) the virtual or digital products in virtual space; and (iii) the data and information links that connect the two [111]. In the field of archaeology, research on Digital Twins is on the rise, such as in graph-based data management for cultural heritage conservation with Digital Twins [112], Digital Twins and 3D documentation using multi-lens photogrammetric approaches [113], and SPARQLing Ogham Digital Twins as Linked Open Data [40] (pp. 119–127), [34] on the basis of the Ogham in 3D project, combining 3D capturing and EPIDOC XML modelling [114,115]. The following subsections describe geometric capturing (Section 2.4.1) and semantic archaeological modelling (Section 2.4.2) in the ARS3D project.

#### 2.4.1. Geometric Capturing

The ARS collection of the LEIZA consists of objects of different sizes. The most miniature objects (pottery shards) have a size of a few centimetres, while the largest objects (plates) have lengths of up to 50 cm. On the objects, there are appliqués with lengths in

the millimetre range. Moreover, the objects are ancient, valuable, and fragile, and must be handled with particular care. For this reason, only non-contact-based acquisition methods are relevant.

Eligible measurement methods must meet several requirements. These include, on the one hand, the necessary flexibility in dealing with objects in the given size categories and, on the other, a guarantee of the necessary accuracy resulting from the requirements of the comparison process, which must take into account the smallest details of ornaments and manufacturer influences. This requirement results in a very high level of precision in the range of 1/10 mm. As a further requirement, the colours of the objects must be represented authentically in the 3D models to esnure that the Digital Twin corresponds to the original as closely as possible.

Various methods can be used for 3D digitisation of small objects. Structured light 3D scanners, hand-held 3D scanners, and digital cameras are suitable for such capture. The highest accuracy required here is achieved by using structured light 3D scanners conceived for industrial applications. Together with digital photogrammetric images, true-to-reality textured 3D models can be generated. In this project, a structured light projection scanner from GOM was used, which is able to capture the geometry of objects with high precision and without contact [116].

Using capture from different perspectives, a 3D point cloud for the entire object is obtained, from which a 3D mesh (3D model) is created after processing. To esnure that 3D models can obtain the appropriate textures, images were taken with a Nikon D800 SLR camera. In order to guarantee colour authenticity, these images were colour calibrated with the help of a color checker. In post-processing, the colour-calibrated images were used to texture the 3D models. Textured 3D models were created in the same way for all ARS objects. These formed the data basis for further processes and analyses. An example is shown in Figure 11.

During data acquisition and processing, technical metadata are generated; these include the capturing device, its settings, and any post-processing steps. These pieces of information are essential, as they provide information on the creation of the 3D models and the quality of the data. In the context of sustainable use according to the FAIR Data Principles, this information is stored in a metadata model. Each 3D model has a file in which the technical metadata are recorded. For 3D data acquisition, a metadata schema and an ontology have been developed for this purpose, in which existing standard designations are included [117].

Overall, this process step provides a precise digital object model close to the original, and together with the qualifying metadata offers the best possible basis for subsequent analyses and sustainable use by external researchers.

### 2.4.2. Semantic Archaeological Modelling

Currently, the status quo in the cultural heritage domain, and as such the "quasi-standard", is relational modelling/RDBMS such as PostgreSQL. Due to the anchoring of references in predefined tables, relational modelling does not have the flexibility and openness required to create a Digital Twin. In contrast, with the help of an ontology, semantic modelling has advantages through graph technologies, such as flexibility and freedom for modelling, openness and transparency for reproducibility, and standardisation through ontologies and (geo-)vocabularies [118]. Semantics allows possible flexibility in the connection of archaeological and geometrical content and the possible application of a (changeable) set of rules that connects geometry and archaeology and provides the basis for conclusions. For open graph-based provision under the FAIR principles, three community standards in particular are used in the field of cultural heritage: the CIDOC Conceptual Reference Model (CIDOC CRM) ontology, the Resource Description Framework (RDF) of the World Wide Web Consortium (W3C), and the Linked Open Data (LOD) Principles [119]. The Open Geospatial Consortium (OGC) standard GeoSPARQL has established itself in the

field of geodata, enabling the semantic description of geodata using simple features and providing functions for querying, such as in PostGIS [120].

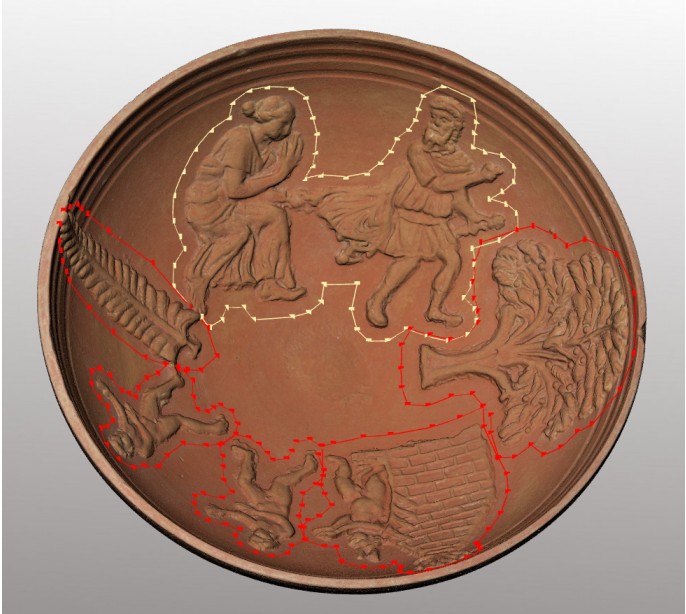

**Figure 11.** O.39675 "bowl with man and woman" in the 3DHOP webviewer, with appliqué man and woman on the top (yellow polygon) (ARS3D project/i3mainz/RGZM, https://creativecommons.org/licenses/by-sa/4.0/ (accessed on 9 March 2023), CC BY-SA 4.0, via https://commons.wikimedia.org/wiki/File:ARS3D_O.39675.png (accessed on 9 March 2023), Wikimedia Commons).

Archaeological information results from interpretation steps in which observations of objects are made and captured features are formed into statements. For example, the top scene on object O.39675 (Figure 11) can be described as follows: a person (woman) is crouching on her knees, touching a person's (man's) coat and dressed in a robe; a person (man/Christ) is bearded, walking to the right, and being pulled/touched on his coat by a person (woman). In order to model these statements, a specialised ARS3D ontology has been designed [121,122]. The ARS3D project uses CIDOC CRM v6.2.1, CRMsci v1.2.3, and CRMdig v3.2.2. These versions were further developed over the last few years, e.g., the term "man-made" was replaced by "human-made" in CIDOC CRM in the last official version (v7.1.2), which is the basis for the initial ISO submission. This statement builds on standards such as the CIDOC CRM reference model, CRM extensions (e.g., CRMinf, CRMsci, and CRMdig), and the Provenance Ontology (PROV-O). In this semantic ARS3D model, objects consist of features described by observations; cf. Figure 12. An African Red Slip Ware object is represented by several features. This information is modelled with CIDOC CRM classes and linked to external references. An object (crm:E24) consists of *n* features (crm:E25) consisting of *n* observations (sci:S4). A feature observation (CRMsci:S4) consists of visual items (again CRMsci:S4) that have activities and/or conditions (crm:E7/E3); cf. Figure 12 (e.g., [123]):

- (obs)-[rdf:type]->(sci:S4_Observation) and (dig:D10_Software_Execution).
- (obs)-[sci:O8_observed]->(obs_detail).
- (obs_detail)-[crm:P2:has_type]->(ars:HumanType) and (ars:man) and ars:woman).
- (obs_detail)-[sci:O8_observed]-> "touched by a woman on his coat"@en (English preference label of a crm:E3_Condition_State instance).
- (obs_detail)-[sci:O8_observed]-> "holding object in left hand"@en (English preference label of a crm:E7_Activity instance).

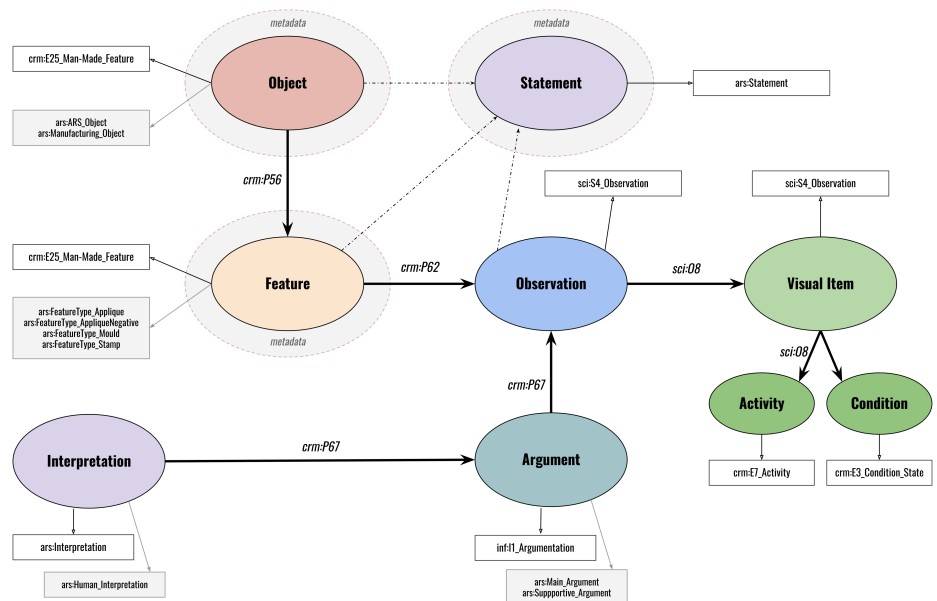

**Figure 12.** Schematic illustration of the ARS3D semantic modelling approach. Objects, features, observations and interpretations are modelled based on CIDOC CRM and its extensions (Observation Modelling Scheme, containing visual items, activities and conditions). (Florian Thiery, https://creativecommons.org/licenses/by/4.0/ (accessed on 9 March 2023), CC BY 4.0).

Each feature is described by a polygon (ars:hasGeometricalExtent); cf. Figure 11. This geometrical annotation as a polygon is the basis for generating a cut feature as well as for the rectification process (Section 2.4).

## 2.5. Individualisation and Comparison of Feature Objects

As the digitised objects may contain several motif elements (called appliqués or features), which are the core of the research, the first required step is to extract them. This individualisation of objects consists of three parts, described in the sections below: visualisation and extraction, rectification, and the comparison of objects.

### 2.5.1. Visualisation and Extraction

In the first step, the appliqués contained in the ARS objects must be isolated and prepared for the comparison processes. The preparation includes archaeological and geometrical steps. From an archaeological point of view, the first step is a visual inspection, including changing the light angle or rotating the object in a viewer, which serves as the basis for the archaeological interpretation concerning fabric (best visible on fractures), vessel wear indications, eventual secondary burning characteristics, etc., and provides the basis for the assignment of semantic content. The visual inspection is connected to the geometric approach, which aims to delineate an appliqué. For this purpose, the appliqué is captured in a 3D viewer using an outline polygon (Figure 13, left) and made available as an independent 3D object via an extraction process (Figure 13, right). The archaeological data are assigned to this object, creating a 3D model enriched with archaeological semantics. In principle, initial archaeological queries can already be made using this model. Geometric queries are not helpful at this point, as the geometry of the carrier object continues to shape the appliqués.

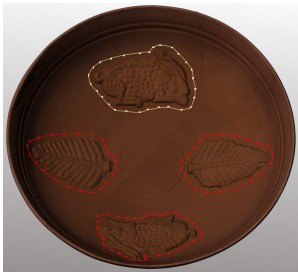 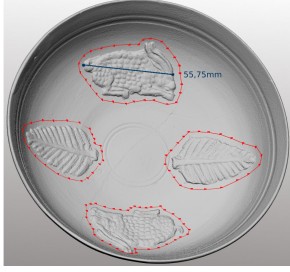 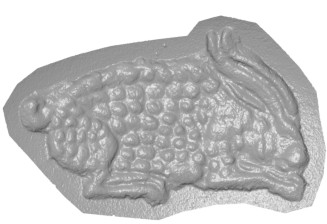

**Figure 13.** (**Left**) Hare appliqué (yellow polygon) on textured object O.39450; (**middle**) marked appliqués (red polygons) on non-textured O.39450 with a measured length of hare (55.75 mm); (**right**) mesh derivative of hare after cutting (ARS3D project/i3mainz/RGZM, https://creativecommons. org/licenses/by-sa/4.0/ (accessed on 9 March 2023), CC BY-SA 4.0).

2.5.2. Rectification

The ARS objects have different vessel curvatures. The appliqués were applied to the carrier objects during manufacturing to adapt to their curvature. Possible shapes range from a plane to an irregularly curved hemisphere. The shapes cannot generally be parameterised, and are sometimes discontinuous. In addition, there may be inaccuracies that occur during the manufacturing process, such as pressure marks or slight distortions due to the handwork. In addition, typical influences on archaeological finds, such as incompleteness, are semantically modelled; however, these object conditions could be extended by annotations concerning wear and material fatigue, which may disturb the original geometry significantly above the 3D scanning resolution of 1/10 mm. However, 3D visualisation tools and analysis methods such as changing light directions or zooming can provide additional information. Nevertheless, the carrier object's influences must be eliminated as much as possible. Only rectified geometries allow a similarity analysis that provides information about the conformity of appliqués.

Rectification Strategy

For the rectification process, a few aspects have to be considered. For example, the area of the appliqué itself should not be included in the estimation of the carrier geometry, as its spatial structure overlays the carrier object. In addition, due to the density of the applied appliqués there is not necessarily an extensive free area available from which a carrier geometry can be estimated. Furthermore, the geometry of the carrier may change in the direct vicinity of the appliqués, which could create false bases for rectification. For this reason, the rectification is limited to the open space near an appliqué. Accordingly, in the course of the extraction from the 3D model, the appliqué is cut out, including a border for geometry estimation (Figure 13).

Most carrier geometries are neither parametrically known nor identical to the existing carrier objects. Most objects are likely to be bowl-shaped while others approximate a plane. As bowl-shaped geometries cannot be unwound without internal shape distortions, rectifying these shapes without error is impossible. Therefore, a residual error must be expected for most objects in the unwinding process.

As previously mentioned, further geometric influences that affect the evaluation of the similarity of appliqués must be taken into account. Thus, a comparison process must be tolerant of specific geometric differences. In the later comparison process, this is considered through corresponding evaluation criteria.

To achieve a high degree of flexibility concerning the most diverse carrier geometries, an iterative rectification procedure is applied, consisting of a best-fit plane with subsequent interpolation of residual errors. Residual fit errors appear for complicated geometries. However, these primarily affect the projection direction (height) and do not significantly change the shape of the projected appliqué, which is of higher importance for comparison. For example, a more significant projection error of 5 mm in the vertical direction affects a shape with a length of 20 mm with only an approximately 0.8 mm positional offset. In the

context of possible wear, spalling, and other influences on archaeological objects, this is acceptable with regard to the overall size of appliqués. In more minor extreme cases, the positional error is significantly lower. For example, a projection error of only 1 mm affects the shape with an error of 0.05 mm. In addition, the observed errors are recorded and are available as an evaluation characteristic for the subsequent steps.

Rectification Process

The process shown in Figure 14 was developed from the above considerations. In this process, the boundary area between the polygon of the circumference and the border of the appliqué is used to calculate a best-fit plane, which then provides the geometric basis for the projection of the appliqué. This best fit results in residual errors that provide information on the quality of the calculation, and thereby the correspondence to a plane. This correspondence is accordingly high for comparatively flat object geometries, while for complex geometries systematic errors arise. The amount of residuals is divided into three categories based on threshold values. Calculations with the lowest residual errors (0.25 mm) fall into category one, and are rated as planar objects, while categories two (<1 mm) and three (>= 1 mm) include slightly and strongly curved geometries, respectively.

For appliqués in these categories, a further correction is made. This correction is guided by the systematic influence in the residuals of the border region (Figure 15A), and applies an interpolation procedure that removes the systematic effects inside the appliqués. The information about the categories is stored for each appliqué (Figure 15B) and is connected to the rectified 2.5D feature (Figure 15C) of each appliqué for later evaluation. In the example shown in Figure 15 there is a mean square residual of 0.7 mm, which is why this appliqué is classified as cat2 and undergoes further correction to remove systematic effects. As can be seen in Figure 15, this interpolation is successful and provides a well-rectified appliqué that provides a good basis for further geometric analyses.

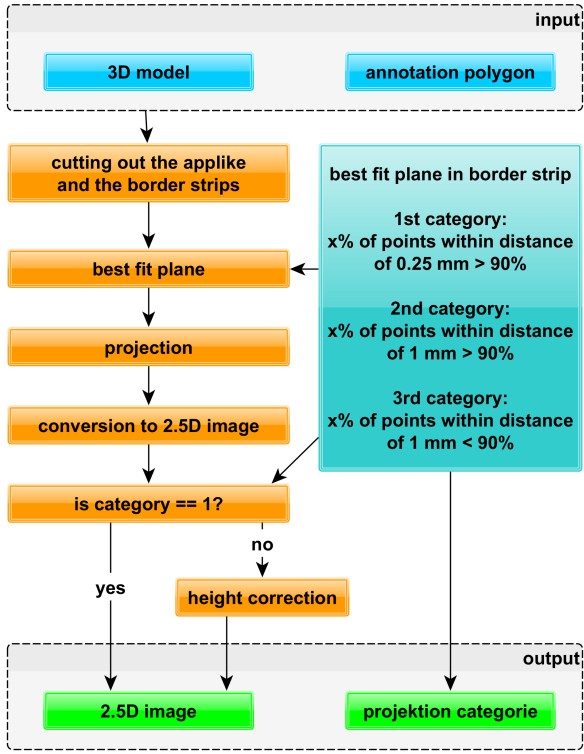

**Figure 14.** Rectification process (Jonas Veller, https://creativecommons.org/licenses/by/4.0/ (accessed on 9 March 2023), CC BY 4.0).

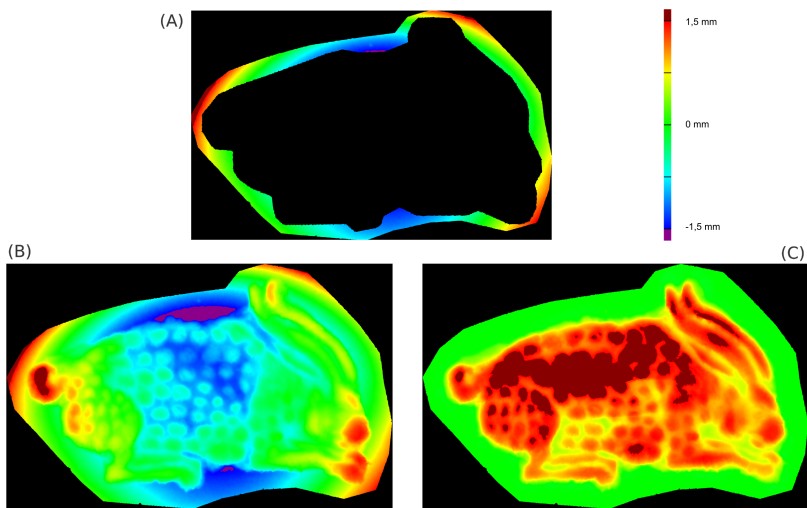

**Figure 15.** (**A**) Residuals of transformation; (**B**) rectified appliqué; (**C**) rectified and corrected appliqué (ARS3D project/i3mainz/RGZM, (accessed on 9 March 2023), CC BY-SA 4.0).

### 2.5.3. Comparison of Feature Objects

By removing the carrier geometry effects, appliqués can now be the subject of similarity analyses. These are based on several steps integrating geometrical and archaeological measures. Ultimately, this provides a rule-based proposition for the similarity hypothesis (Figure 16).

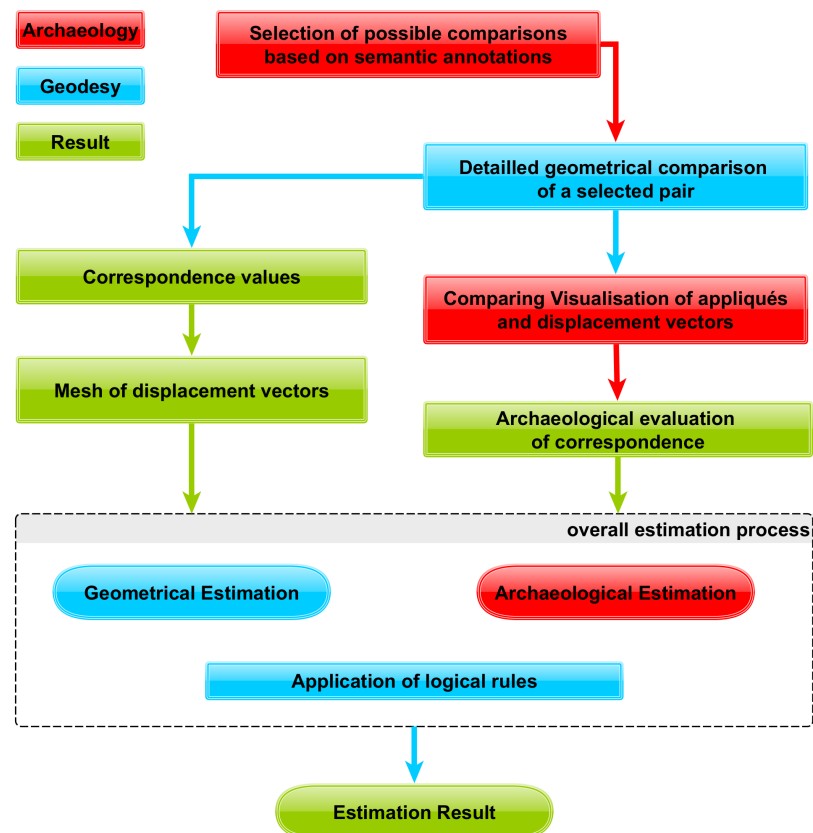

**Figure 16.** Representation of the entire comparison process workflow (Jonas Veller, https://creativecommons.org/licenses/by/4.0/ (accessed on 9 March 2023), CC BY 4.0).

Geometrical Comparison of Objects

Comparison aims to provide information about global or local geometric matches and support archaeologists in evaluating items from their professional points of view. Due to archaeological assumptions about the manufacturing process and the possible influences of the previous calculation steps, various scenarios are possible:

1.  Two appliqués are identical and match completely down to the last detail;
2.  Two appliqués are identical but do not match everywhere due to local changes;
3.  Two appliqués are partially identical but do not match in many places;
4.  Two appliqués have a similar basic geometry but hardly match in the details;
5.  Two appliqués are disturbed by geometric influences and only match locally in a few places;
6.  Two appliqués differ completely in shape and/or size despite having the same archaeological annotation.

These scenarios result in several demands on a comparison procedure. On the one hand, the global alignment of the appliqués to be compared must be provided automatically. On the other hand, it must be possible to make a statement that evaluates even the most minor surface parts. In addition, the necessary robustness against imperfect rectifications or defects in the objects must be provided. Because the derivatives are available as 2.5D objects, image processing methods can be applied for this comparison (Figure 17). In principle, feature- or area-based methods are both possible [124]. Because of the relatively smooth surfaces, however, feature techniques are less suitable. In part, this is because results must be available over the entire object surface; feature-based methods cannot guarantee this, as they depend on the occurrence of features. Therefore, the area-based multi-stage procedure described below has been developed in order to derive the necessary information for global alignment and local matching from templates of different sizes.

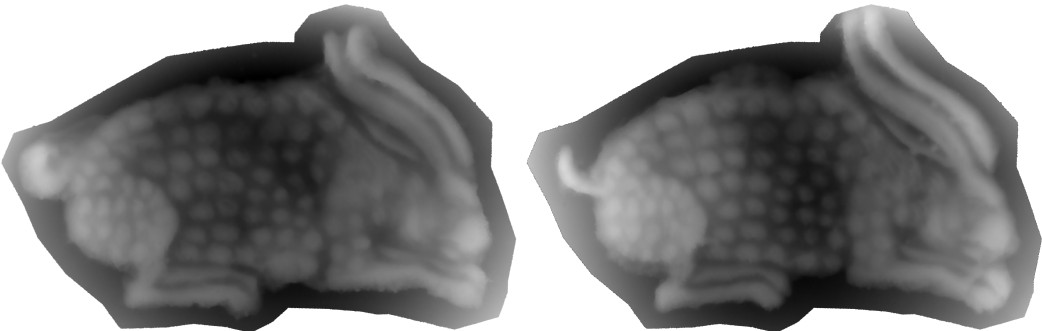

**Figure 17.** Grey values scaled on an 8-bit image on bowl with hare and branch O.39450; cf. Figure 13. (**Left**) "Hare 1", stated as Armstrong 6.170, described in [19] (pp. 149–150); (**right**) "Hare 2", stated as Armstrong 6.169, described in [19] (p. 149) (ARS3D project/i3mainz/RGZM, https://creativecommons.org/licenses/by-sa/4.0/ (accessed on 9 March 2023), CC BY-SA 4.0).

Process

The process outlined in Figure 18 is based on an iterative procedure that moves from global calculations to local ones, adjusting the amount of object information considered in each case. Each stage works with a grid of anchor points at which matches are made. The density of the grids increases with the degree of refinement, and in the most detailed level with a spacing of approximately 1 mm, provides a sufficient basis for determining local correspondences. The position of the grid points in the target image is determined using the transformation results from the previous stage. A manually determined approximate value for the rotation between the appliqués is used for the first iteration. In the first iteration, large windows are used to ensure that sufficient information is available and that disturbing influences on the calculation are less dominant due to the rotation not yet being precisely known. Each matching provides the necessary information from the

difference between the start and target positions to improve the transformation between the appliqués. Considering the ambiguities that occur due to repetitions in the surface or homogeneous areas, e.g., due to regular patterns, support measures must be taken into account. These ambiguities apply to blunders as well, which can occur individually or in more significant numbers. Accordingly, a maximum permissible distance between the start and end position of a target is defined for each iteration. This method prevents drift of matches when repeating patterns or the generation of implausible results in the case of weak morphological content. Furthermore, an outlier analysis is carried out in each stage, which relates individual results to those of the neighbourhood. This analysis allows the detection of implausible results, and prevents their negative effects on improvement of the transformation. The final result of the geometrical comparison is a multi-level dataset with the residuals from the different iteration steps and the transformation parameters. These data points express the geometric view of a presumptive match, and are one component in the final evaluation. Interpretation from an archaeological point of view can then extend this information.

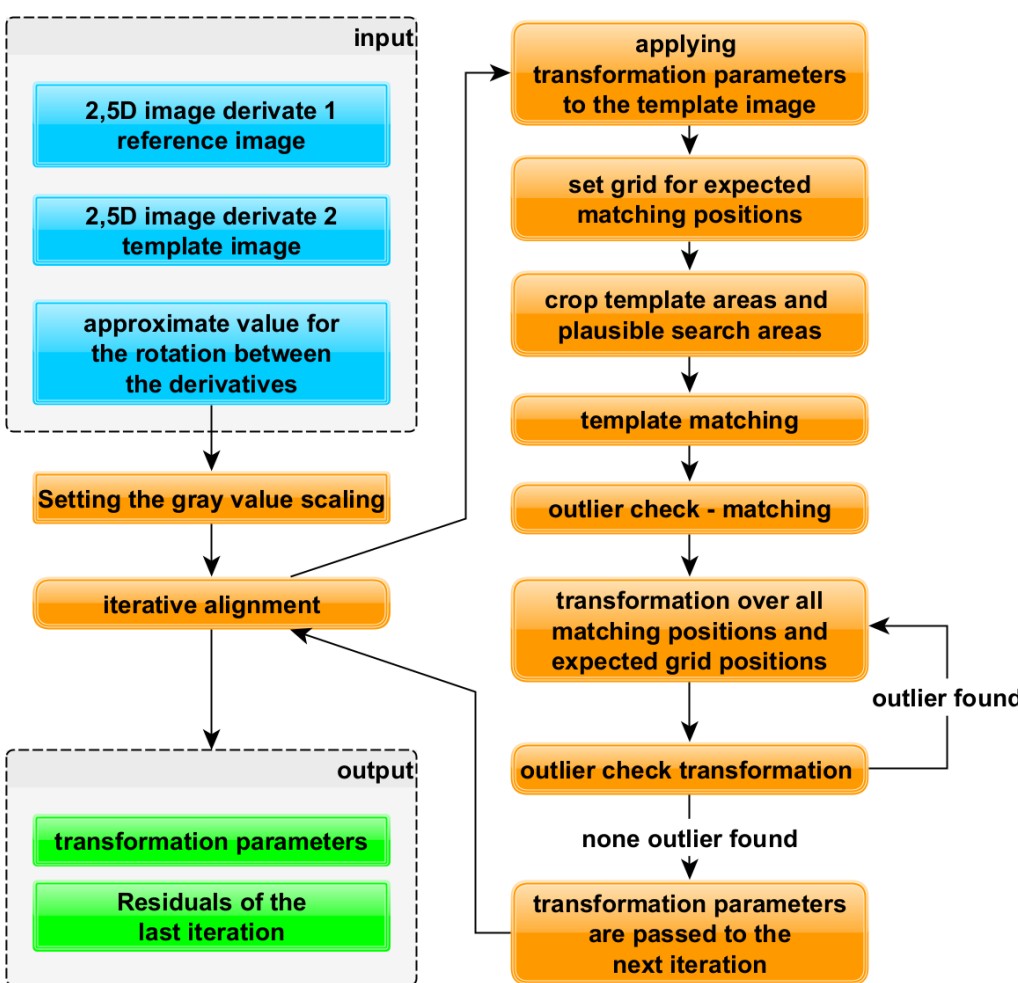

**Figure 18.** Scheme of the geometric alignment process (Jonas Veller, https://creativecommons.org/licenses/by/4.0/ (accessed on 9 March 2023), CC BY 4.0).

Visual Comparison of Appliqués

The geometric correspondence of two appliqués cannot be the sole basis for a complete similarity assessment. For example, the manufacturing process and influences from the use of the objects affect the evaluation. Objects may be essentially identical while differing in significant details. Alternatively, appliqués may show major changes in parts, e.g., an arm has a different position due to the process of application to the carrier object. In the former

case, while the geometric correspondence is high, the objects are classified as different. In the latter case, it is the opposite. In this respect, an archaeological evaluation of the comparison result is indispensable. For this purpose, a viewer has been developed that allows one to view the results in various ways, providing the basis for archaeological evaluation.

Visualisation uses two geometrically coupled windows in which the compared appliqués can be displayed and analysed in detail with true-to-scale parallel pan and zoom functions (Figure 19). Because each window has different layers, various other visualisations are possible [125]. For example, both applications can be overlaid in one viewer and viewed simultaneously by changing the transparency. In another variant, displacement vectors from the highest detail level of the geometrical comparison can be superimposed on the appliqués. These vectors allow local effects to be viewed and assessed. To support interpretation, the quality of the respective displacements associated with the correlation (r) is colour-coded (green: r > 80%, yellow: 60% < r < 80%, red: r < 60%). While vectors tend to indicate a shift in position, the superimposition of the appliqués with the height differences between them allows for assessment in the vertical direction. For this purpose, the height difference image can be loaded into a layer and analysed together with the appliqué. If all options are taken together, archaeologists have many display variants at their disposal. With the help of these options, a detailed analysis of the geometric correspondence and the results of the calculation process can be made and used as a basis for a professional evaluation of the comparison.

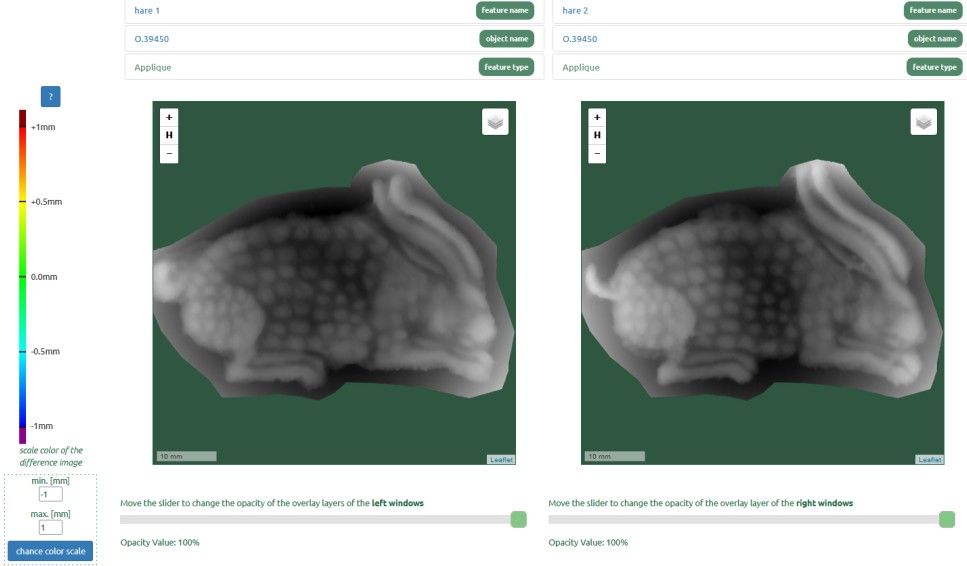

**Figure 19.** Comparison viewer with example comparison: "O.39450 hare 1 hare 2". (ARS3D project/i3mainz/RGZM, https://creativecommons.org/licenses/by-sa/4.0/ (accessed on 9 March 2023), CC BY-SA 4.0).

Archaeological View of the Features Comparison

To compare features, archaeological research and the application of Semantic Modelling are necessary. In our case, archaeologists can use different figure types to distinguish appliqués and obtain a first impression, such as the different hares described by Armstrong 6.163–6.179 [19] (pp. 148–151). Taking the example of the two hares on" bowl with hare and branch O.39450" (Figure 13, left, top: hare 1, bottom: hare 2), these can be identified as figure types by Armstrong [19]; hare 1 (Figure 17, left) can be identified as Armstrong 6.170 [19] (pp. 149–150) and hare 2 (Figure 17, right) as Armstrong 6.169 [19] (p. 149). Armstrong described figure type 6.169 as "Hare crouching right, with its ears curled up at the ends, resting its head on its front paws" [19] (p. 149), and described 6.170 as the same, adding "It is almost identical to 6.169, except for its round tail" [19] (p. 150). Additionally,

these hares can be described using terms of iconographical thesauri, such as IconClass "25F26(HARE)" (rodents: hare) (https://iconclass.org/25F26(HARE), accessed on 9 March 2023); Getty AAT "300250216" (hares (Leporidae) (http://vocab.getty.edu/aat/300250216, accessed on 9 March 2023) is a general term referring to any of more than 30 species belonging to the genus Lepus); or Wikidata "Q46076" (Lepus, genus of mammals) (http://www.wikidata.org/entity/Q46076, accessed on 9 March 2023).

Rules-Based Assessment of Similarity Hypothesis

The aim of the described geometrical evaluation process is to create a residual-based argumentation aid for archaeologists. For this purpose, the geometrical process automatically provides an individual assessment of the similarity of the compared appliqués. This estimation, combined with an archaeological decision rule set, builds the basis for comparing feature objects.

The automatic evaluation of the geometric comparison results in a score between 0 (different) and 3 (equal). The determination of the score is based on comparison results for the different matching levels. For similar appliqués, it can be assumed that a large correspondence (score: 3) will result for both coarse and fine grids, while unequal appliqués consistently achieve low correspondences, which are expressed in low correlations and large remaining residuals (score: 0). However, appliqués may only partially match, for example, if there is a local defect. In order to evaluate these correctly, further analysis of the residuals must be carried out. This analysis examines subgroups of the grid points and checks whether only a smaller group of residuals has a larger RMS (root mean square) value. This happens, for example, in the case of local defects. Depending on the statistics, these are marked with 1/2 in the score (borderline case) [59] (pp. 5–6), [126].

The archaeological comparison (Figure 20 based on [127] (p. 632) uses both the detailed geometrical data and the calculated geometrical score. It is supported by visualisations within the comparison viewer. Generally, local and global differences or similarities are distinguished. If local or global conspicuousness is observed, an evaluation of the reason follows, e.g., damage, modifications during the manufacturing process, or additional material residues. Depending on possible reasons and the extension of the modification, a rule-based estimation, that is, an archaeological score of the similarity of two features, is performed, e.g., manual modification = 0, wear of damage = 1, no conspicuity = 3. The minimum local and global scores are combined into a minimum archaeological score, which leads to the determination of whether the features are different or equal. However, an archaeological veto, combined with a detailed explanation, may be applied if the professional observation is discrepant from the calculated estimation [59] (p. 7), [128].

The results of both comparison processes are combined into a final average score (3: equal, 1–2: borderline case, 0: different), resulting in three comparison categories: equal, borderline, and different (Figure 21) [127] (p. 633). This score is calculated as the weighted average of the archaeological and geometrical estimation scores. This combined score is set within ranges that form the conclusion: x > 1.9: equal; >0.9x <= 1.9: borderline; and x <= 0.9: different [59] (p. 7), [126].

Regarding the hare feature comparison mentioned above, the applied calculation based on the rule set results in a "borderline case" (https://ars3d.rgzm.de/comparisonviewer/index.htm?comparison_id=68b95623-9c15-44ad-b7b3-e494f0205c32, accessed on 9 March 2023) of an overall estimation with a minimum average score value of 1.0, which is close to "different". The geometrical estimation identified an interference and set the geometrical score to 1. The archaeological estimation identified local conspicuity (shift or twist) and set the archaeological score to 1. These can be seen both on the different tails from [19] (p. 150) and on the stretching of the hares along the horizontal axis.

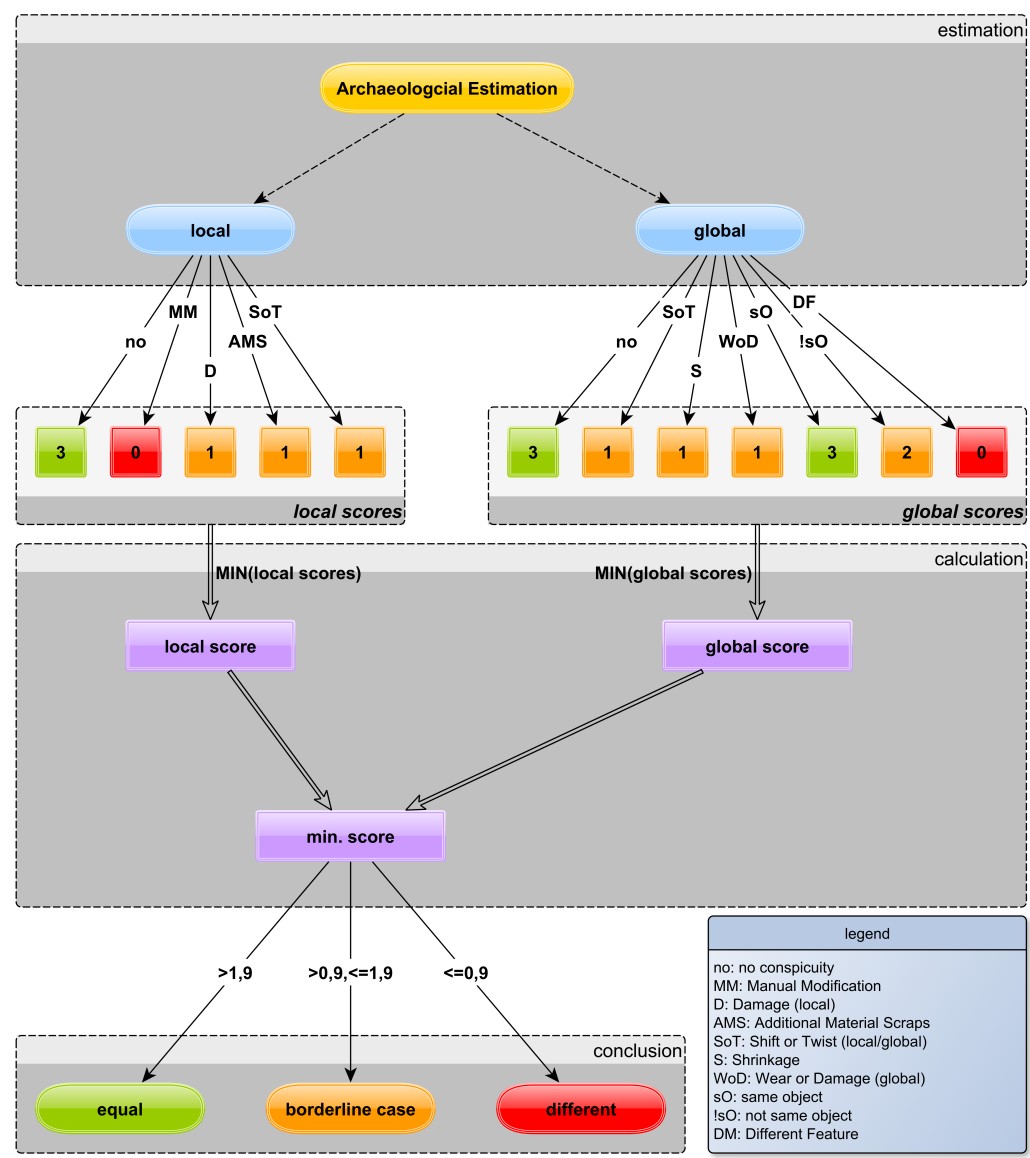

**Figure 20.** Schematic illustration of the calculation and ruleset of the archaeological estimation, based on [127] (Figure 4) (Florian Thiery, https://creativecommons.org/licenses/by/4.0/ (accessed on 9 March 2023), CC BY 4.0).

## 2.6. Semantic Modelling of Comparisons

The results are semantically modelled as RDFs to achieve FAIR data and provide the comparison conclusions in a comprehensible way with the goal of reproducible research. The semantic modelling of comparisons is part of the ARS3D Ontology [121] using prefixes; c.f. Table 1. Comparison follows the rules shown in Figure 22. The transformation into an RDF file [129] is carried out using a Python script [126]. The comparison results are modelled as an interpretation based on two arguments, namely, a geometrical and an archaeological argument. These arguments are described in detail using observations. Interpretations, arguments, and observations are related via CRM property P67 (refers to); cf. Figure 22.

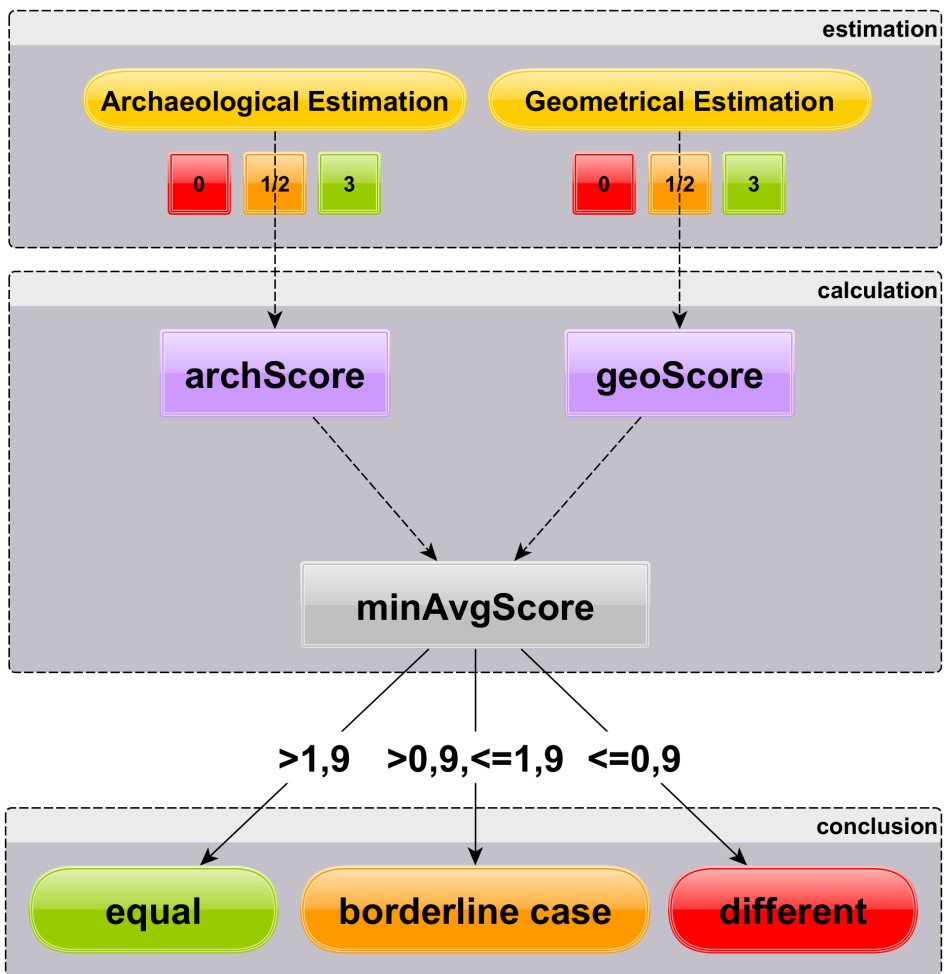

**Figure 21.** Schematic illustration of the calculation and ruleset for a comparison estimation, based on [127] (Figure 5) (Florian Thiery, https://creativecommons.org/licenses/by/4.0/ (accessed on 9 March 2023), CC BY 4.0).

**Table 1.** Prefixes.

| Prefix | Uri |
| --- | --- |
| ars: | http://ars3d.rgzm.de/ontology# |
| rdfs: | http://www.w3.org/2000/01/rdf-schema# |
| prov: | http://www.w3.org/ns/prov# |
| crm: | http://www.cidoc-crm.org/cidoc-crm/ |
| inf: | http://www.ics.forth.gr/isl/CRMinf/ |
| sci: | http://www.ics.forth.gr/isl/CRMsci/ |
| dig: | http://www.ics.forth.gr/isl/CRMdig/ |

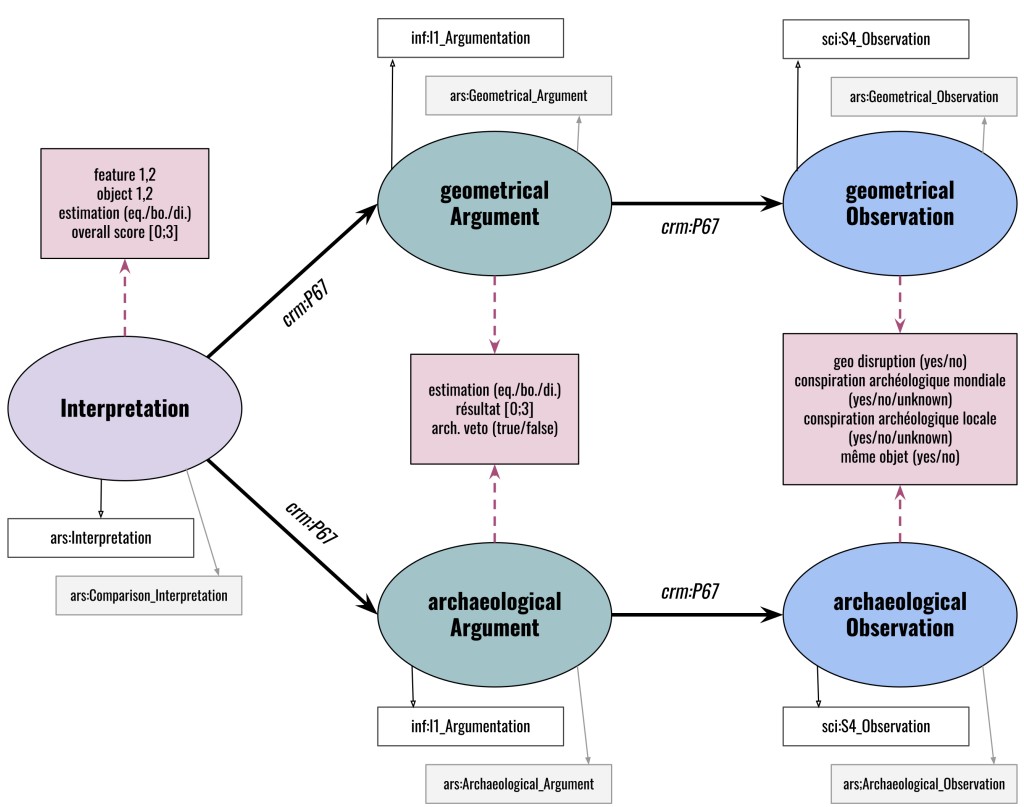

**Figure 22.** Schematic illustration of semantic comparison modelling (Florian Thiery, https://creativecommons.org/licenses/by/4.0/ (accessed on 9 March 2023), CC BY 4.0).

The overall result is modelled as an ars:Interpretation with the type ars:Comparison_Interpretation. In addition to the metadata (rdfs:label, operator (dig:L30), date (dig:L31), thumbnail, and provenance information for scripting using PROV-O), each interpretation is related to two features (ars:comparison_feature1/2) and one or two objects (ars:comparison_object1/2), and contains the overall ars:estimation (ars:Equal, ars:BorderlineCase, ars:Different) as well as the ars:score value [0;3]. The overall estimation result and the resulting score as the average of the archaeological and geometrical minimum score are calculated using the rules described in Section 2.5.3. Each ars:Comparison_Interpretation refers to two arguments (inf:I1_Argumentation), an ars:Geometrical_Argument and an ars:Archaeological_Argument. The arguments are described with an ars:estimation and a minimum ars:score. The archaeological argument can contain a veto against the implemented logic result (ars:archaeologcial Veto), which can be expressed as text (ars:archaeologcialVetoDescription). The estimation and score are calculated using the rules described in Section 2.5.3. Furthermore, any argument refers to an observation (sci:S4_Observation), an ars:Geometrical_Observation, and an ars:Archaeological_Observation. On the one hand, the geometrical observation describes disruptions (ars:geometricalDisruption) with ars:Yes or ars:No; on the other, the archaeological observation has three characteristic properties: ars:sameObject (ars:Yes or ars:No); ars:globalConspicuity (ars:Yes or ars:No), if yes, which kind of ars:globalConspicuity Type (ars:ShiftOrTwistGlobal, ars:Shrinkage, ars:WearOrDamageGlobal, ars: DifferentFeature); and ars:localConspicuity (ars:Yes or ars:No), if yes, which kind of ars:localConspicuityType (ars:ShiftOrTwistLocal, ars:DamageLocal, ars:AdditionalMaterialScraps, ars: ManualModification). These properties are the basis for the comparison logic rule set described in Section 2.5.3.

## 3. Results

The following results of the processing chain discuss selected examples. Based on the available dataset of 336 objects with a total number of 414 partially or entirely preserved

features, a total number of 469 comparisons are possible. From these comparisons, 88 are categorized as equal, 216 as borderline, and 165 as no correspondence.

### 3.1. Identical Woman and Boy Appliqués

An example of two appliqués identified as identical can be seen in Figure 23A,B. It shows two "woman and boy" figure types on fragmented tableware. This "woman and boy" is part of the "Thetis entrusts Achilles to Chiron" interpretation, described in Armstrong 9.3A-D [19] (p. 234) "Achilles and Priam and Achilles Cycle" (Q111385205) and S.V. Löwenstein as "Die Übergabe Achills an Chiron" [20] (p. 433) (Q111385245). The scene shows Thetis (woman: goddess of water, daughter of the ancient sea god Nereus) walking towards the centaur Chiron and leading Achilles (boy: son of the Nereid Thetis and Peleus, king of Phthia), who is walking behind her and holding her hand, right [20] (pp. 433–434). In the profile view (Figure 23C), the predominantly flat geometry of the carrier object can be seen. Only a weak bend is visible towards the edges of the support surface, meaning that the conditions for correcting the vessel geometry are good. Different preconditions result from the smaller edge area for object O.40531, which means that less geometric information is available to estimate the geometry. As seen in the profile view, the appliqués are located at the edge of the unrolled part of the carrier object, which affects the area recorded for the estimation of the carrier geometry and introduces a slight discontinuity into the geometry estimation. To a lesser extent, this applies to the geometric edge visible at the lower edge of the woman's garment as well.

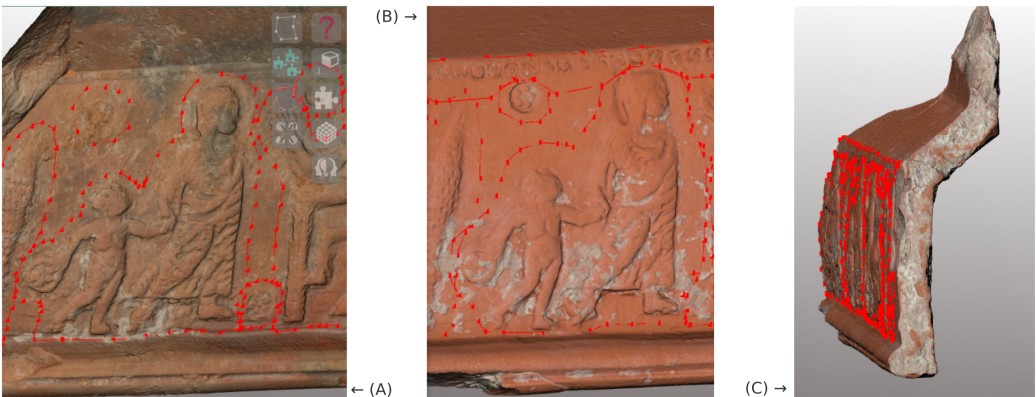

**Figure 23.** Appliqué "woman and boy" on the objects O.40531 (**A**) and O.42129 (**B**); (**C**) profile view from object O.42129 (ARS3D project/i3mainz/RGZM, https://creativecommons.org/licenses/by-sa/4.0/ (accessed on 9 March 2023), CC BY-SA 4.0, via https://commons.wikimedia.org/wiki/File:ARS3D_O.40531_O.42129.png (accessed on 9 March 2023), Wikimedia Commons).

A first visual estimation of the two appliqués suggests a high degree of similarity and a cross-fade video [130] (Part I) (Video/Figure 24). Only different ageing processes can be assumed from the differences in visual appearance. In addition to the colour differences, smaller deviations in the morphological details indicate the same.

If we look at the result of the geometric correction for the appliqué on object O.42129 (Figure 23a), a slight influence of discontinuities in the woman's head and the edge of her robe can be seen. However, these only affect the height, and not the shape, which is essential for a geometric comparison. All contours remain undisturbed. This result applies to the second appliqué as well, which is not shown here.

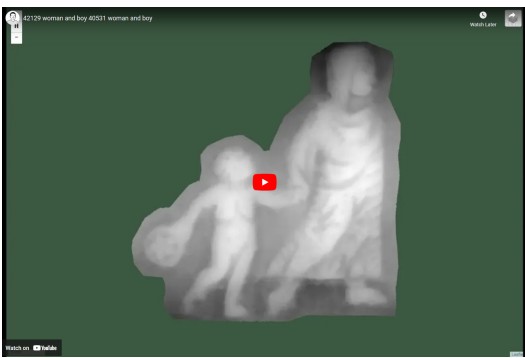

**Figure 24.** Video of 42129 woman and boy 40531 woman and boy by [130] (Part I) available at https://youtu.be/ktc2hkQRon0 (accessed on 9 March 2023), (Florian Thiery and Jonas Veller, https://creativecommons.org/licenses/by-sa/4.0/ (accessed on 9 March 2023), CC BY-SA 4.0).

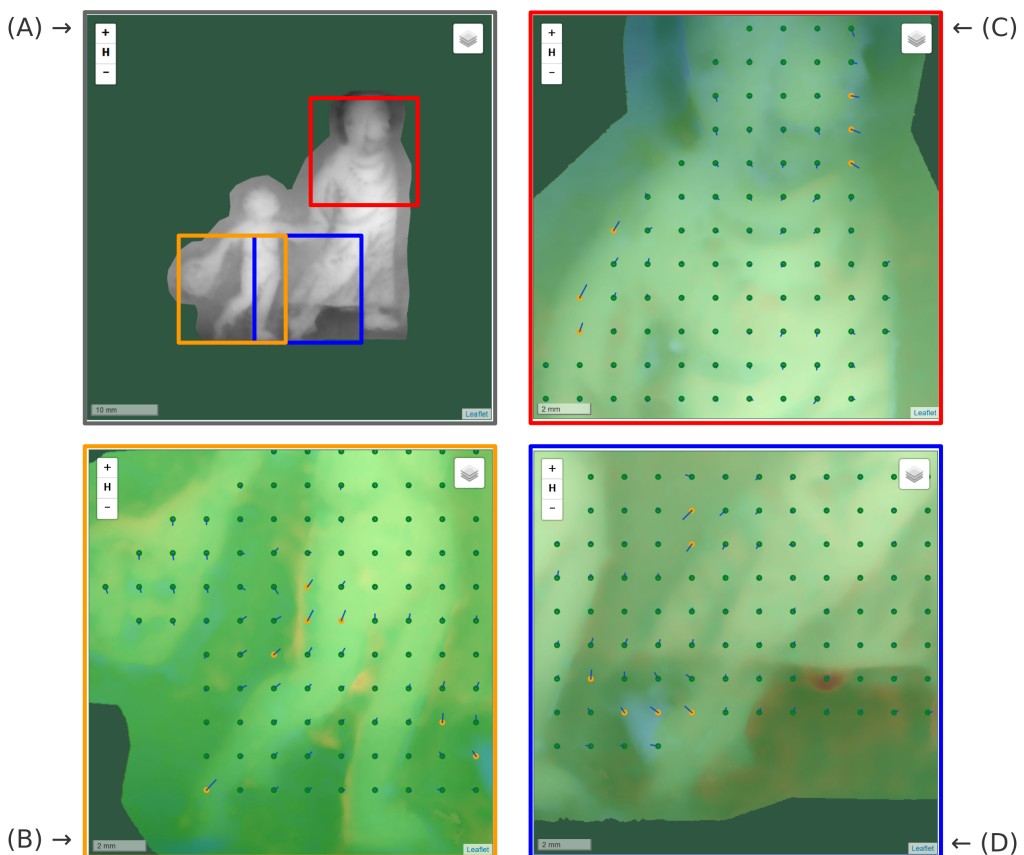

**Figure 25.** (**A**) Object O.42129 rectified; (**B**) child subsection with overlaid difference image and residuals; (**C**) head subsection with overlaid difference image and residuals; (**D**) foot subsection with overlaid difference image and residuals (ARS3D project/i3mainz/RGZM, https://creativecommons.org/licenses/by-sa/4.0/ (accessed on 9 March 2023), CC BY-SA 4.0).

The results of the geometry comparison can be seen in Figure 25b–d. They show the superposition of several components of the comparison. The rectification of appliqué O.42129 shown in Figure 25a is chosen as the lowest level. At the next level, it is overlaid with the colour-coded height difference image and in the uppermost level by the residual image of the determined shifts in shape between the two appliqués.

First, the significant geometric correspondence between the two appliqués can be seen in the consistently green colour of the different images. On the one hand, this correspondence is supported by the high similarity in the local comparison process (green circles), and on the

other by the consistently low displacements (short vectors). Only in a few marginal areas are somewhat more significant shifts visible (yellow circles) (Figure 25b–d). These appear in areas that could have been slightly displaced, for example, by the application of the appliqués to the carrier vessel (boy's leg (b), woman's arm (c), and woman's foot (d)). As these are small and only locally occurring differences, the overall evaluation remains unaffected from a geometrical and archaeological perspective. Even the visual differences (Figure 23a) do not contradict the possibility that the appliqués match, as they do not affect the local calculation and as such can be attributed to minor age-related influences.

Correspondingly, the semantic model proposes to understand the appliqués as identical (https://ars3d.rgzm.de/comparisonviewer/index.htm?comparison_id=83a33d18-660e-43b2-b0e2-2629e86f01fc, accessed on 9 March 2023). The geometrical view provides a score of 3; the archaeological interpretation raises only a very few doubts, and provides a score of 2. This results in a score above the threshold for matching appliqués (>1.9).

### 3.2. Damaged Human–Horse Hybrid

The object "dish with boat and hybrids" (O.40534) displays four features: one boat and three human–horse hybrids characterised with a "mantle" (class clothing) observation, as well as "looking to the left" and "standing" activity. These feature applications are not mentioned in the relevant literature as having a deeper iconographic or mythical background. However, they are made available to the research community via Wikidata as Q111370381 (ARS3D G/FT I [77] (pp. 5–6)) and Q111372221 (ARS3D M/FT I [77] (pp. 12–14)), as well as by the ARS3D project [77].

This example shows a comparison that does not result in a clear match. The object can be seen in Figure 26, and shows so-called human–horse hybrids. In the present case, the appliqués lie on the outer edge of a dish. Thus, there is slight curvature of the carrier geometry in the longitudinal and transverse direction of the motif. In addition, the carrier object is broken, causing a disturbance in the lower area of the suitable appliqué (horse-hybrid-2).

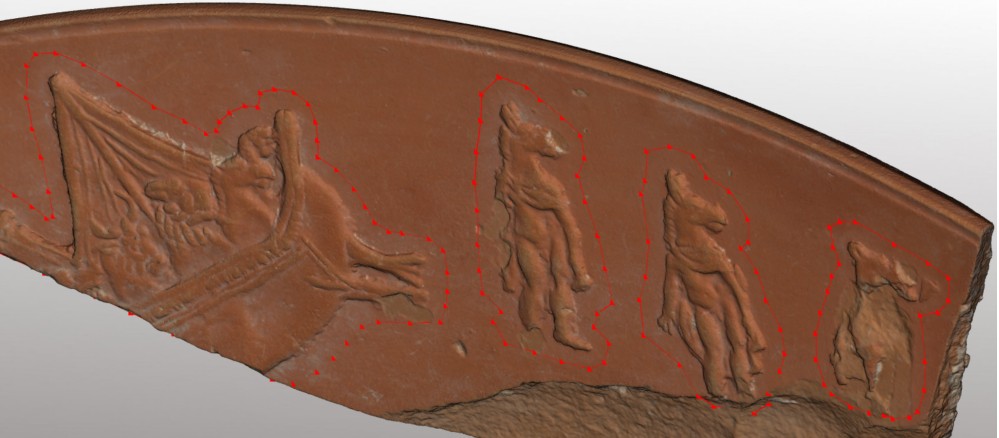

**Figure 26.** Scene "dish with boat and hybrids" on object O.40534 with four appliqués (ARS3D project/i3mainz/RGZM, https://creativecommons.org/licenses/by-sa/4.0/ (accessed on 9 March 2023), CC BY-SA 4.0, via https://commons.wikimedia.org/wiki/File:ARS3D_O.40534.png (accessed on 9 March 2023), Wikimedia Commons).

The purely visual comparison shows two similar motifs in their basic structure, which could indicate a common origin during manufacture. However, it can by no means be taken for granted that these appliqués stem from one single appliqué mould (thus, from the same workshop), as appliqués may stem from a series of very similar-looking appliqués made in one single mould [78] (p. 45, Abb. 11). Each impression in such a mould for the serial production of appliqués may differ, e.g., due to different impression angles. Directly visible differences result from damage that may have been caused by the use of the bowl

or may be linked to the part that has broken off. In addition, differences may have been produced while applying the appliqué to the carrier object. This must be judged for each individual case. In general, it can be stated that if the applied appliqués show traces of, e.g., clay blurring continued on the vessel itself, the distortion can only be explained by the process of applying the appliqué.

Figure 27 shows the two rectified appliqués. The projection has operated correctly. Only in the area of the disturbance caused by the break in the dish can specific height influences be seen; however, these do not influence further comparison. In this respect, the prerequisites for a meaningful comparison are present.

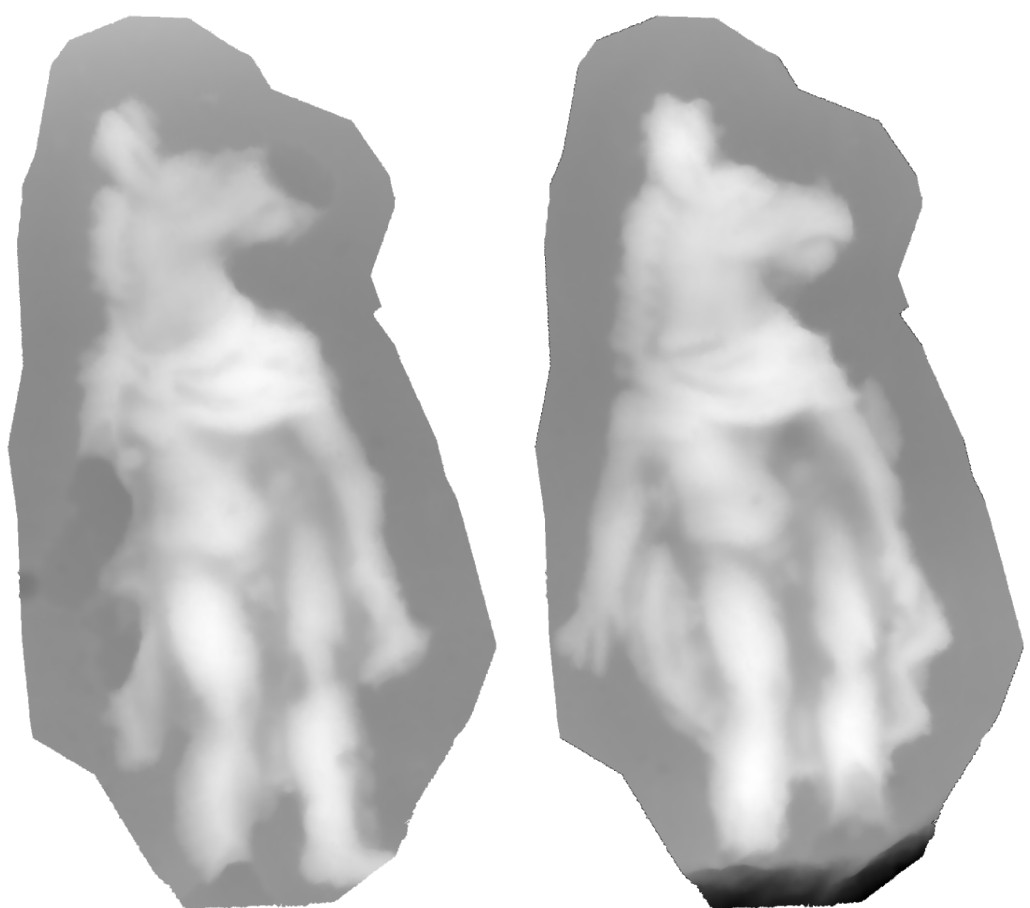

**Figure 27.** Appliqué "human–horse hybrid" No. 1 (**left**) and No. 2 (**right**) of object O.40534 (ARS3D project/i3mainz/RGZM, https://creativecommons.org/licenses/by-sa/4.0/ (accessed on 9 March 2023), CC BY-SA 4.0).

The first impression of the fundamental similarity between the motifs and a cross-fade video [130] (Part II) (Video/Figure 28) is confirmed by considering the local content. Thus, Figure 29 shows the head part of the motif and Figure 30 shows the torso. There are apparent similarities visible in both areas. For example, the horse heads show similar components (neck, mane, ears, head). These similarities apply to the rest of the body (legs, arms, torso) as well. The differences result from posture or damage. For example, the left horse's head is missing a part of its nose, while the left arm is missing from the trunk. In addition, the left horse's head is turned backwards, and slight twists are visible in the torso. This conspicuity is especially visible in the displacement vectors. In this respect (Figure 31), the two appliqués could have undergone changes caused by the process of application or resulting from later use (flaking).

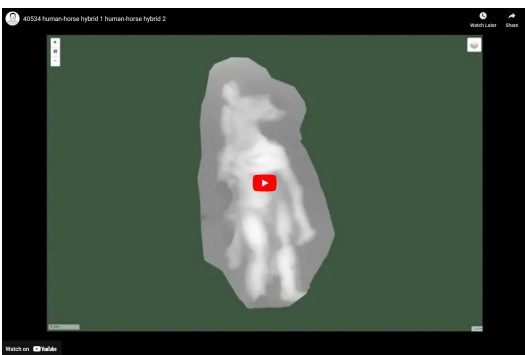

**Figure 28.** Video of 40534 human–horse hybrid 1 and human–horse hybrid 2 by [130] (Part II) available at https://youtu.be/wEB5dg3sBwU, (accessed on 9 March 2023), (Florian Thiery and Jonas Veller, https://creativecommons.org/licenses/by-sa/4.0/ (accessed on 9 March 2023), CC BY-SA 4.0).

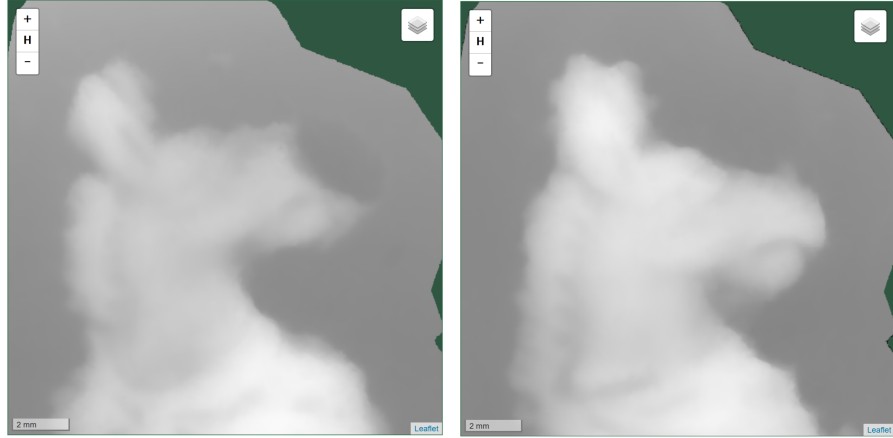

**Figure 29.** Appliqué "human–horse hybrid" No. 1 (**left**) and No. 2 (**right**) of object O.40534: head part (ARS3D project/i3mainz/RGZM, https://creativecommons.org/licenses/by-sa/4.0/ (accessed on 9 March 2023), CC BY-SA 4.0).

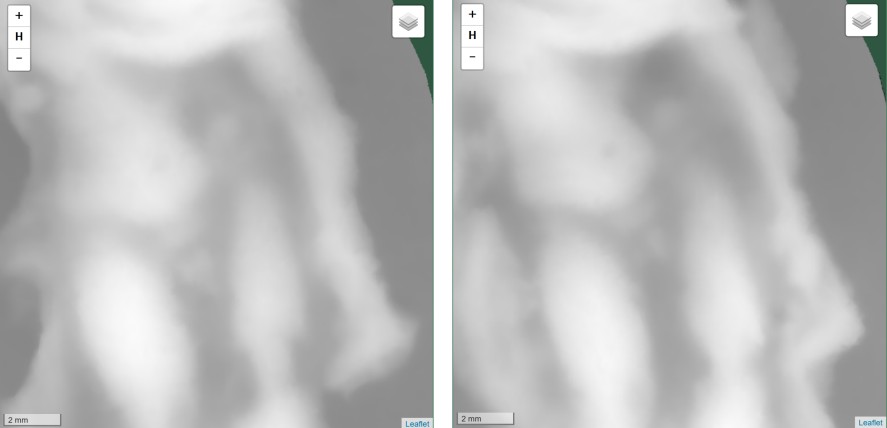

**Figure 30.** Appliqué "human–horse hybrid" No. 1 (**left**) and No. 2 (**right**) of object O.40534; torso (ARS3D project/i3mainz/RGZM, https://creativecommons.org/licenses/by-sa/4.0/ (accessed on 9 March 2023), CC BY-SA 4.0).

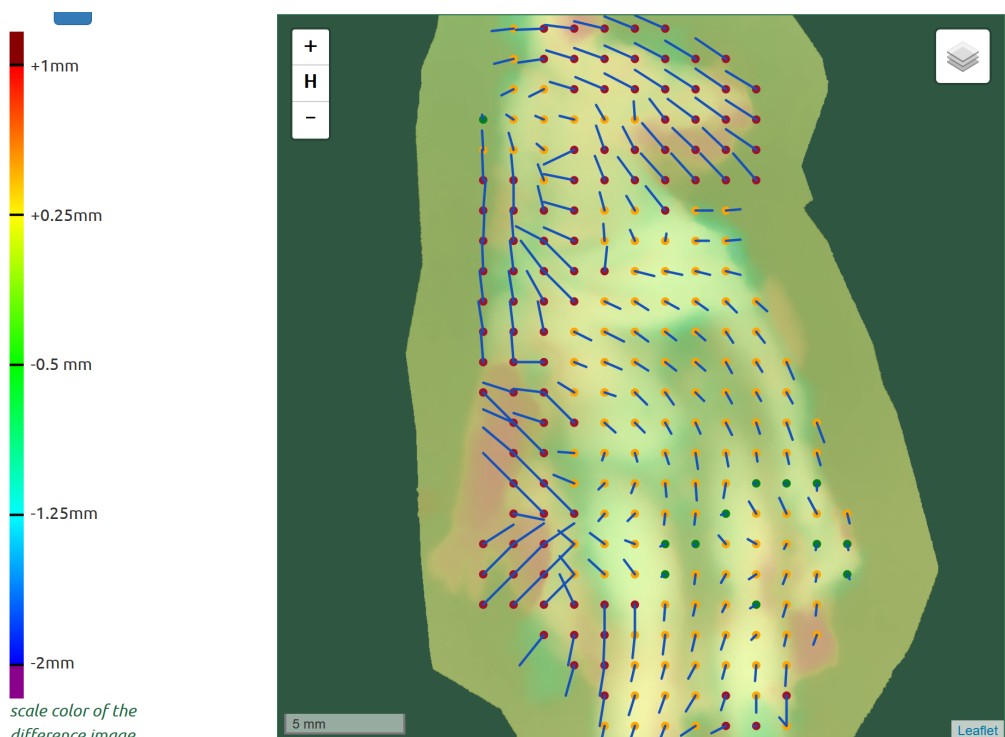

**Figure 31.** Appliqué "human–horse hybrid 1" of object O.40534 with overlaid residuals (ARS3D project/i3mainz/RGZM, https://creativecommons.org/licenses/by-sa/4.0/ (accessed on 9 March 2023), CC BY-SA 4.0).

However, the semantic model classifies the resulting situation as borderline. The geometric and the archaeological evaluation contribute equally to this (a score of 1 in each case). From a geometrical point of view, the quality of the similarity values is not explicitly high. In addition, the displacements in the motif are comparatively large. From an archaeological point of view, doubts arise about the motif's defects and changes.

### 3.3. Bear with Local Twist/Shift

A semantic search for "bears" through observations of object features shows the following result: bears with features as appliqués (positive) that are made from clay: count 13. These are described by statements by Armstrong 6.74 [19] (p. 135) (Q111385273) "bear leaping right" (3), 6.82 [19] (p. 136) (Q111385281) "bear standing" (2), and 6.87 [19] (p. 137) (Q111385288) "bear leaping left". The bear after Armstrong 6.74 appears on objects O.40761 (twice) and O.39599 "bowl with animals". This example (40761_bear_1_bear_2) compares the two bears on O.40761 "bowl with animal hunt".

This last example shows a comparison between motifs applied to the surface of a large bowl. To a certain extent, they document the limits of the current process chain. This ambiguity appears mainly due to the bowl's solid geometric over-shaping of the motifs. The appliqués extend along the circumferential direction and show strong curvature influences in their respective longitudinal and transverse directions. The size of the appliqués contributes to this effect (Figure 32).

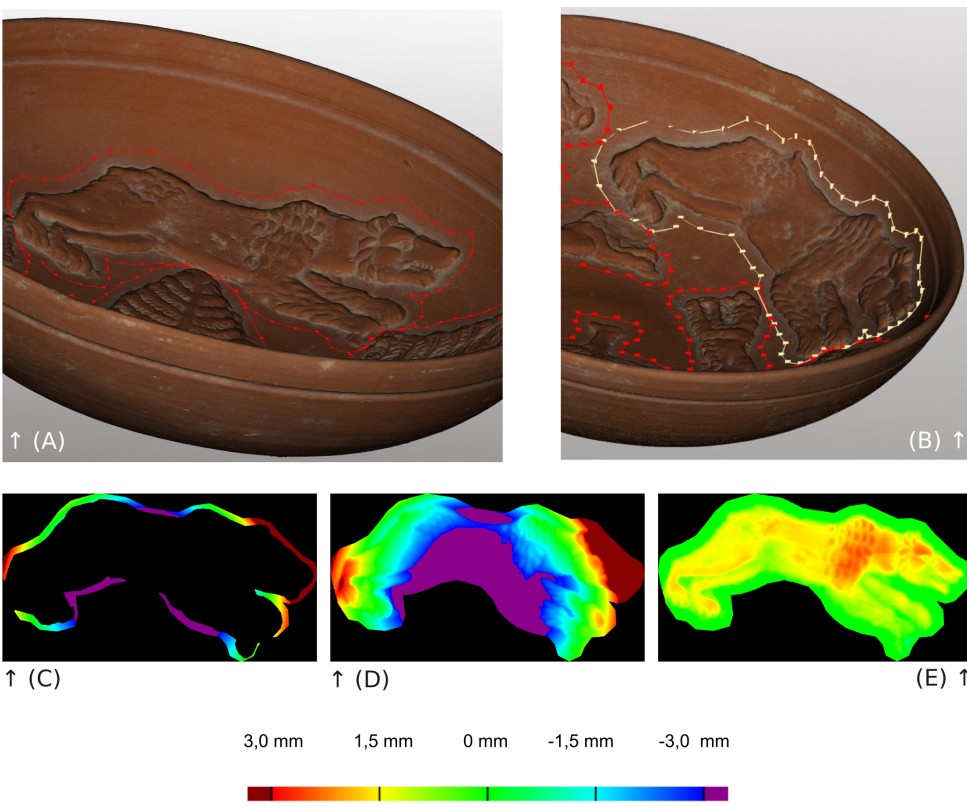

**Figure 32.** Appliqués of object O.40761: (**A**) appliqué "bear 1"; (**B**) appliqué "bear 2"; (**C**) systematic effects after rectification of bear 1; (**D**) bear 1 before correction; (**E**) bear 1 after correction (ARS3D project/i3mainz/RGZM, https://creativecommons.org/licenses/by-sa/4.0/ (accessed on 9 March 2023), CC BY-SA 4.0).

Examining the basic structure of the motifs, a similarity can be assumed here. Both motifs show jumping bears, which are similar in posture and in the shape of the head, torso, and legs. This similarity is additionally showcased in the cross-fade video [130] (Part III) (Video/Figure 33). The presumption of similarity is accordingly quite high (Figure 34A,B).

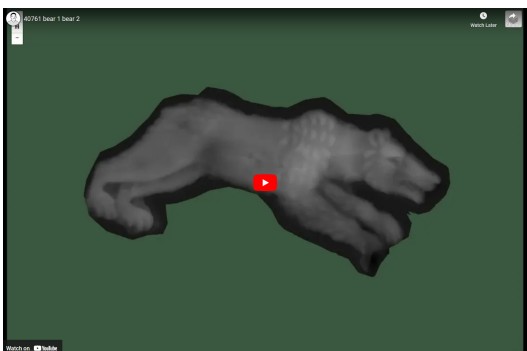

**Figure 33.** Video of 40761 bear 1 and bear 2 by [130] (Part III), available at https://youtu. be/mthmDcvEpIw (accessed on 9 March 2023), (Florian Thiery and Jonas Veller, https:// creativecommons.org/licenses/by-sa/4.0/ (accessed on 9 March 2023), CC BY-SA 4.0).

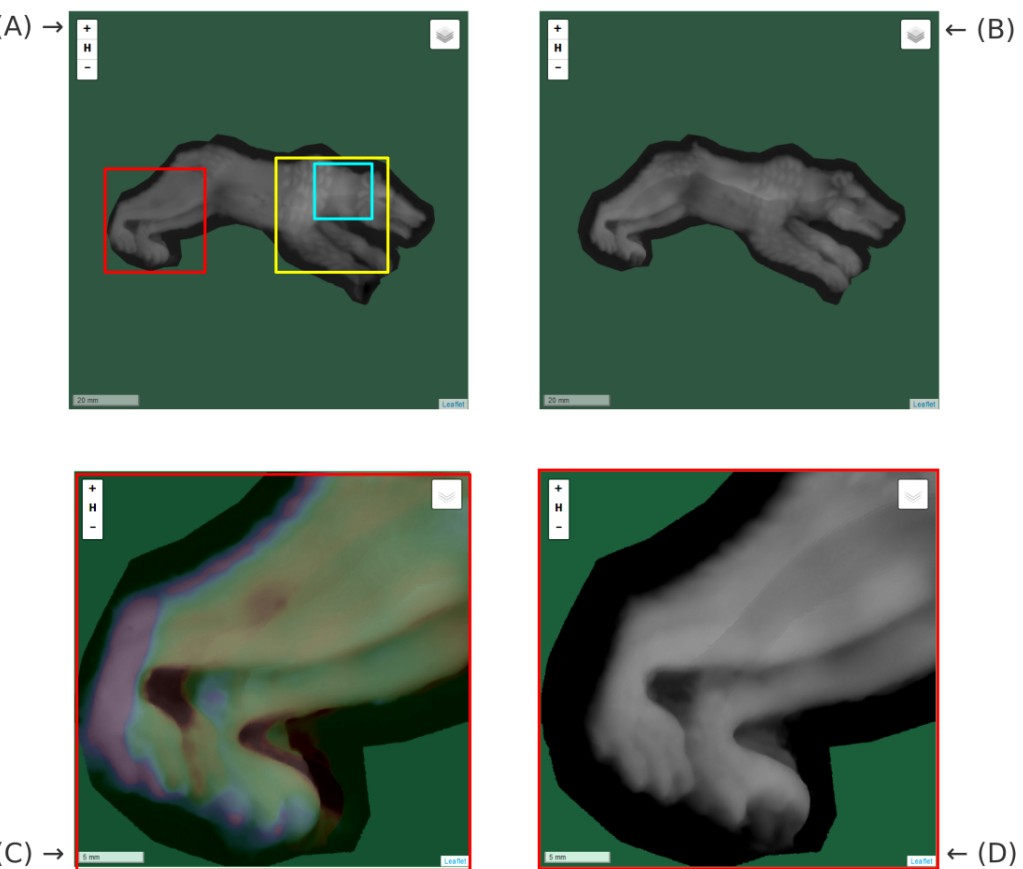

**Figure 34.** Appliqué "bear 1" (**A**) and "bear 2" (**B**) of object O.40761. (**A**) shows different frames of the bear in red, yellow and cyan. Overlaid height differences of appliqué "bear 1" (**C**) and "bear 2" (**D**) of object O.40761 (red frame) (ARS3D project/i3mainz/RGZM, https://creativecommons.org/licenses/by-sa/4.0/ (accessed on 9 March 2023), CC BY-SA 4.0).

However, unwinding the basic geometry of a shell results in systematic errors. These errors are already visible in the results of the rectification process (Figure 32C), which is why the correction is classified with the lowest quality (rms > 1 mm). Accordingly, the residual errors from the projection are large and systematic, and the projection itself does not provide a useful geometry (Figure 32D). Instead, we obtain geometrically useful datasets after compensation for the systematic effects (Figure 32E), showing that this correction works even in more challenging cases. However, artefacts can be identified along the longitudinal axis of the bear (Figure 34B). This affects the similarity comparison, which is why the results of the geometric comparison have to be marked as interfered (score: 0).

Looking at further details, we see differences and similarities. The back legs are displaced between the two motifs, as can be seen in the height differences in Figure 34C. In addition, the displacement vectors (Figure 35, left) indicate movement within the motifs. To a certain extent this might be due to a rectification error; however, because of the size of the offset and the fact that such an effect only occurs in this area, it may influence the production process. Because of a very local distortion or twist (e.g., the legs of a bear appear more stretched [130] (Part III)), the archaeological estimation is a "borderline case", and it is awarded a score of 1.

Similarities are found on the front parts of the appliqués. As the displacement vectors show (Figure 35, middle), the torso, forelegs, and head largely match except for a moderate displacement. An area with uncertain results is found on the neck. However, the subject has little structure, as seen in the contrast-enhanced image (Figure 35, right). Accordingly, the matchings are uncertain. Similarly, the low structure in the area of the back legs contributes

to the lower similarity. Therefore, the shifts, which are uniform in themselves, are connected with more significant uncertainty. Accordingly, the different threshold values in the geometry assessment evaluate the results as uncertain, resulting in a score of 0.

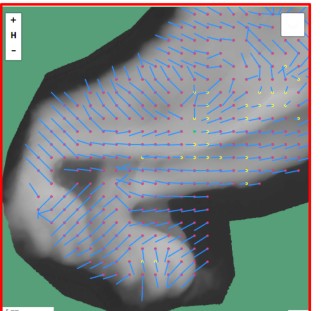 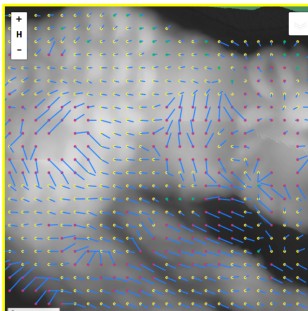 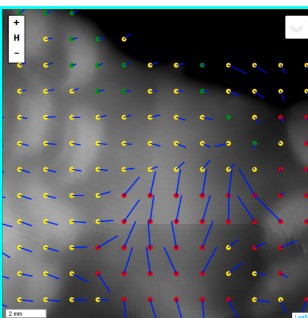

**Figure 35.** Overlaid residuals of appliqué "bear 1"; the colours of the clips correspond to Figure 34A. (**Left**) Back leg (red frame); (**middle**) upper body, front legs, and head (yellow frame); (**right**) neck (contrast enhanced) of object O.40761 (cyan frame) (ARS3D project/i3mainz/RGZM, https://creativecommons.org/licenses/by-sa/4.0/ (accessed on 9 March 2023), CC BY-SA 4.0).

Consequently, the semantic model produces the result "different", as too many interfering influences exist, preventing judgment as to the underlying similarity between corresponding features with the required quality.

## 4. Discussion

The presented workflow is fully realised and provides a functioning working environment for archaeological research via the ARS3D portal (https://ars3d.rgzm.de, accessed on 9 March 2023). The complexity of the solution naturally offers several starting points for discussion.

Considering the geometrical aspects, the first question is whether the digital models allow for such detailed geometrical comparisons. The answer to this question is clearly in the affirmative. The close-to-the-original models are recorded with the highest precision and represent an optimal basis for an analysis from both a geometric and visual point of view. Moreover, they provide an excellent basis for visually supported interpretation. Only haptic or physical observations are impossible here. However, these limitations are negligible compared to the benefits of making digital objects available to the global research community.

Concerning the correction of the geometric influence of the carrier objects onto the appliqués, residual influences remain in the case of complex geometries, which is why the method has room for improvement. However, there is no simple solution, as the geometry to be estimated is hidden by the appliqués. At the same time, the surfaces may be neither unwindable nor continuous. The area size necessary for estimating more complex geometries is not available, which is why using the developed approximation method seems reasonable. It works well with the small amount of surface information at the border of the appliqués, and is robust against possible overfitting. Only the interpolation logic for transferring the determined systematic influences to the surface of the appliqué offers room for improvement in the case of large objects. Artefacts that can occur at large distances from the border could be reduced in this way.

On the other hand, the determined residual defects are documented and evaluated from a geometric point of view. Thus, they are available as objective features for further analyses, and are included, for example, in the set of rules applied in the evaluation of similarities. Suppose that one considers the extent of any remaining correction errors concerning the room for interpretation from an archaeological point of view applied to other changes in appliqués (traces of use, damage, craft influences). In this case, the latter contains more significant uncertainties. In this respect, minor effects due to rectification are tolerable and do not pose a problem in a process that is fundamentally marked by remaining uncertainties. In this sense, the observed

amount of 28% of comparisons resulting in a conclusion of "unequal" does not indicate a significant impact of possible errors on the geometric correction process.

In addition to the numerical statements of the comparison, the latter is based on a practical and flexible visualisation of the results, which enables a correct understanding. For this purpose, the differences determined in the comparison are shown both as horizontal shifts and as vertical differences. This makes the similarities and differences in the appliqués visible and interpretable. The simultaneous visualisation of both compared appliqués and the geometric coupling of the representations additionally supports interpretation by the user. This makes the comparison results comprehensible for archaeologist users and provides the basis of obtaining an assessment from their professional point of view. This assessment then enters into the set of rules, allowing an objective and comprehensible overall assessment that simultaneously provides an entry point into the professional discourse between users.

The archaeological comparison process covers many possibilities for the description of characteristics that allow conclusions related to the complex issue of figure stamp origins, including: (1) if appliqués are not entirely identical, but stem from the same patrix (as the observed distortion is explicable by the original vessel curvature), the same origin can be assumed; (2) a figure type may be a surmoulage; (3) a figure stamp is different because a part of the applied figure stamp is missing. Nevertheless, not all possibilities are covered because of the small corpus of the LEIZA ARS collection and the limitation that only features are compared, rather than the mould and the possibly resulting appliqué. The collection of the LEIZA does not contain any moulds and appliqués that have been used together. The inclusion and analyses of further collections could close this gap and extend the comparison with a "real match". Due to this, the semantic rule set and the possibilities have to be extended, and an archaeological veto against the rules must be formalised.

The geometric comparison processes may only indicate which areas of an appliqué or an object need to be looked at in greater detail. However, the comparison of multiple appliqués showing the same motifs allows the study of recurring distortion patterns. This distortion pattern enables inferences related to, e.g., discernable distortions along the longitudinal axis that appear to coincide with the observation that these figure stamps stem from the most curved part of a vessel. Conversely, the analysis of a rectificated figure stamp would even allow for suggestions as to where a figure stamp was positioned on the original vessel.

Several different steps are taken to make the ARS3D data and comparison results comprehensible and reusable in the sense of FAIR data [131] (Table 1): semantic (meta)data modelling based on community standards is performed, e.g., CIDOC CRM, PROV-O, SKOS (RDA-R1.3-02D/RDA-R1.3-02M); the data are embedded into the Linked Open Data Cloud and the Wikidata Knowledge Graph (RDA-I3-01D/RDA-I3-04M); meta(data) are accessible through free access protocols on Zenodo (RDA-A1.1-01D RDA-A1.1-01M); and the data are identified by a globally unique identifier, especially URIs (RDA-F1-02D).

The rules are currently not modelled in a machine-readable ontology; rather, they are applied in a Google SpreadSheet and Python scripts [126]. ARS3D data are published as Linked Open Data [132] (https://github.com/RGZM/ars-lod, accessed on 9 March 2023) stored in an RDF4J triplestore, which enables the provision of a SPARQL endpoint in the archaeology.link LOD hub (http://archaeology.link, accessed on 9 March 2023) as well as embedding into Wikidata [133] using bi-directional links, which offers great potential for reuse. This linking uses the same match property P2888 in Wikidata and the exact match property from the Linked Archaeological Data Ontology (LADO). In the future, the creation of an archaeology.link property within Wikidata, similar to, for example, ToposText within a community discussion (https://www.wikidata.org/wiki/Wikidata:Property_proposal/ToposText_IDs, accessed on 9 March 2023), could create greater sustainability with the help of the Wikidata community.

## 5. Conclusions

To summarise the results presented above, an overall functional process digitises a whole genre of ARS objects with close-to-the-original quality, extracts individual appliqués, processes them geometrically, checks them for geometric correspondence with other appliqués, links the geometric content with specialist archaeological content, stores it in a higher-level semantic model, and visualises it in various ways, thereby making it available for interaction with scientists and allowing it to be queried in a web environment. The process has been applied to the whole collection, and shows a highly probable similarity rate for 21% of objects, whereas 51% have less probable similarities and only 28% have unreliable or contradictory correspondences. These results confirm the assumption that the production process must lead to similarities in appliqués and that this should be visible when comparing them. Together with the archaeological content, this provides a basis for considering further research questions. To ensure that research on this collection is as broadly based as possible, all content is openly accessible and modelled based on the latest methods of linked data management. Moreover, the archaeological data are linked to the digital object models and the geometric derivatives. As a result, all content relevant to research is available to all those who wish to use it for their own research purposes.

The methodological approach using semantically modelled decision trees presented in this study fundamentally enhances previous ARS research. The combined comparisons of appliqués by geometrical and archaeological observations and statements allow future ARS research to present verifiable and comprehensible insights into the relations between ARS potters and potteries. Our methodological approach provides ARS research with a multiperspective view beyond the current art-historical iconographical and iconological figure stamp comparisons.

**Author Contributions:** Florian Thiery: Research Software Engineer, Linked Open Data and Semantics Expert and Geodesist (Conceptualization, Formal analysis (archaeological semantics, comparison analysis), Investigation (archaeological semantics, comparison analysis), Methodology, Software (editor, portal, semantics, databases, APIs, ontology, comparison and LOD transformation scripts), Visualization (portal and editor), Writing—original draft); Jonas Veller: Research Software Engineer and Geodesist (Conceptualization, Formal analysis (comparison analysis), Investigation (3D digitalisation, processing derivates, comparison analysis), Methodology, Software (3D-HOP, comparison process, comparison viewer), Visualization (comparison viewer and derivates), Writing—original draft); Laura Raddatz: Geodesist (Conceptualization, Investigation (3D digitalisation, processing derivates, comparison analysis), Methodology, Software (Post Processing Digitisation, metadata scripts), Visualization (3D models), Writing—original draft); Louise Rokohl: Archaeologist (Conceptualization, Formal analysis (archaeological semantics, comparison analysis), Investigation (archaeological semantics, comparison analysis), Methodology, Writing—original draft); Frank Boochs: Professor of Geodesy and Applied Informatics at HS Mainz (Conceptualization, Formal analysis (comparison analysis), Funding acquisition, Investigation (comparison analysis), Methodology, Supervision, Writing—original draft); Allard W. Mees: Archaeologist and Head of Department of Scientific IT at LEIZA (Conceptualization, Formal analysis (archaeological semantics), Funding acquisition, Investigation (archaeological semantics, comparison analysis), Supervision, Writing—review & editing). All authors have read and agreed to the published version of the manuscript.

**Funding:** The African red slip ware in digital form—3D documentation for the multi-perspective analysis of a central object category of the late antiquity period Project (ARS3D) was funded by the Federal Ministry of Education and Research Germany (BMBF), Förderkennzeichen: BMBF-01UG1888AX, BMBF-01UG1888BX.

**Institutional Review Board Statement:** Not applicable.

**Informed Consent Statement:** Not applicable.

**Data Availability Statement:** Data described in this paper are available at OSF (https://doi.org/10.17605/OSF.IO/6HJ7G), Zenodo (https://zenodo.org/communities/ars3d, accessed on 9 March 2023), and on Wikidata, in particular, the comparison data (https://doi.org/10.5281/zenodo.5647827), comparison scripts (https://doi.org/10.5281/zenodo.5647864), graph data (https://doi.org/10.5281/zenodo.5642750), ontology (https://doi.org/10.5281/zenodo.5642891), images of features (https:

//doi.org/10.5281/zenodo.5645236), comparisons (https://doi.org/10.5281/zenodo.5645253), and the source code of the web application (https://doi.org/10.17605/OSF.IO/P5TKW) and the API (https://doi.org/10.17605/OSF.IO/WRJ7K).

**Acknowledgments:** We would like to thank all former and student project members and the BMBF for funding. Especially the student assistants Samantha Beck and Anke Dingler for adding the archaeological semantics and annotations. Special thanks go to Songül Polat, Lisa Patel, Daniel Walker, and David G. Wigg-Wolf for proofreading as well as to Jörg Drauschke and Benjamin Fourlas for their provenance research related to the LEIZA ARS collection. Moreover, we are thankful for Latex and Overleaf support by Sophie C. Schmidt and Martina Trognitz, for 3D scanning support by Anja Cramer and Guido Heinz, and for project application support by Stefanie Wefers.

**Conflicts of Interest:** The authors declare no conflict of interest.

## Abbreviations

The following abbreviations are used in this manuscript:

| | |
|---|---|
| AI | Artificial Intelligence |
| AMT | Academic Meta-Tool |
| ARS | African Red Slipware |
| ARS3D | African Red Slip Ware Digital |
| ARSW | African Red Slip Ware |
| BMBF | German Federal Ministry of Education and Research |
| CAA | Computer Applications and Quantitative Methods in Archaeology |
| CAD Model | Constructed Design Model |
| CARE | Collective Benefit, Authority to Control, Responsibility, Ethics |
| CC BY | Creative Commons Licence Author Attribution |
| CC0 | Creative Commons Licence 0 (Public Domain) |
| CH | Cultural Heritage |
| CIDOC | Comité International pour la Documentation |
| CIDOC CRM | CIDOC Conceptual Reference Model |
| CMS | Corpus of Minoan and Mycenean Seals |
| CRSW | Cypriot Red Slip Ware |
| DL | Deep Learning |
| ERSW | Egyptian Red Slip Ware |
| FAIR | Findable, Accessible, Interoperable, Re-usable |
| Getty AAT | Getty Art and Architecture Thesaurus |
| HTTP URI | Uniform Resource Identifier for an object on the World Wide Web |
| i3mainz | Institute for Spatial Information and Surveying Technology at HS Mainz |
| LADO | Linked Archaeological Data Ontology |
| LEIZA | Leibniz-Zentrum für Archäologie |
| LOD | Linked Open Data |
| LRA | Late Roman A |
| LRB | Late Roman B |
| ML | Machine Learning |
| OGC | Open Geospatial Consortium |
| PROV-O | Provenance Ontology |
| PRSW | Phocaean Red Slip Ware |
| QID | Wikidata Q-Identifier |
| RDA | Research Data Alliance |
| RDF | Resource Description Framework |
| RDF | Reference Description Framework |
| RGZM | Römisch-Germanisches Zentralmuseum |
| RSE | Research Software Engineer |
| SKOS | Simple Knowledge Organization System |
| SPARQL | SPARQL Protocol And RDF Query Language |
| SR | Semantic Reasoning |
| SRSW | Sagalassos Red Slip Ware |

| URI | Uniform Resource Identifier |
| W3C | World Wide Web Consortium |

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
