# Peer review of "A Semi-Automatic Semantic-Model-Based Comparison Workflow for Archaeological Features on Roman Ceramics"

_ijgi, doi:10.3390/ijgi12040167_

Round 1
Reviewer 1 Report (Previous Reviewer 2)
-
Author Response
Dear Reviewer 1, many thanks for your constructive criticism and supportive contributions during the reviewing process.
Reviewer 2 Report (Previous Reviewer 3)
Once reviewed the paper and read the comments made by the authors to my previous assessment, I consider that the paper has been successfully improved and its weak points were properly amended. Therefore, this new version of the paper can be published in its current form.
Author Response
Dear Reviewer 2, many thanks for your constructive criticism and supportive contributions during the reviewing process.
Reviewer 3 Report (New Reviewer)
Some digital platforms for classifying Roman amphorae or fine wares have existed for quite some time – admittedly not with sufficient detail, interactivity, and user friendliness. Most examples provide visual reference groups, with or without associated provenance information, and therefore basically transfer book-based classifications to a website with some sort of search engine. This current paper wishes to move beyond simple taxonomy, by applying AI analyses to ARS.
To be fair, other publications in the last two or three years have focused on digitizing and simulation-aided artificial intelligence of red slip wares. Some were created as apps or other forms of software, while others are more conceptual, but they all share some type of image acquisition, followed by an AI processing stage. The statement of AI applications on Roman pottery being rare may be correct in absolute terms, but there has been a growing number of outputs in the last 24 months or so, which will likely only increase in the immediate future.
What seems to be new in this study is the scope of the data, on the one hand, and the reevaluation of classical methodological questions on typological micro-provenance or detailed mold and appliqué chronology, for instance. It is in a comparative context that this exercise becomes interesting, as it allows for the proper interpretation of very small distortions between equal decorative elements.
In short, this is a fine paper that might be reviewed in terms of textual coherence. Here and there it seems some sentences were added perhaps by different co-authors, and on multiple occasions a sort of non-English native orality shines through, but this is mainly a formal aspect that does not affect the fundamental quality of the research itself. I suggest rereading the entire paper with a fresh pair of eyes. If I may add another recommendation: a short paragraph in the conclusions, stating the added value of the findings in very straightforward terms (along the lines of “before AI we had X, now we have Y”), might be very useful. Most readers will acknowledge both the inherent and explicit potential of this research, but summarizing it in one or two crystal clear, practical sentences would surely benefit the paper.
Author Response
Dear Reviewer 3, many thanks for your constructive criticism and supportive contributions during the reviewing process. Our project started in 2018 and indeed since then, further AI-orientated technologies and applications in archaeology have been developed. But it remains a rare category. Particularly at the beginning and during the project, we evaluated the state-of-the-art technologies and libraries fitting our purposes and following that we developed the procedures and methodology described in the study.
Our paper has undergone English language editing by MDPI and is now certified by their author services. The text has been checked for correct use of grammar and common technical terms by an experienced, native English-speaking editor.
We added the following paragraph at the end of the conclusion section:
“The methodological approach using semantically modelled decision trees presented in this study fundamentally enhances previous ARS research. The combined comparisons of appliqués by geometrical and archaeological observations and statements allow future ARS research to present verifiable and comprehensible insights into the relations between ARS potters and potteries. Our methodological approach provides ARS research with a multiperspective view beyond the current art historical iconographical and iconological figure stamp comparisons.”

This manuscript is a resubmission of an earlier submission. The following is a list of the peer review reports and author responses from that submission.
Round 1
Reviewer 1 Report
This article covers many very interesting topics.
However, it seems to me that it tries to cover too many aspects and therefore dilutes the message and makes it less effective.
The introduction gives a very broad picture of the evolution of archaeological practices - a bit like a position paper - and figures 1, 2 and 3 are very nice but are they really necessary? I wonder about the first sentence of the article. Are we in the middle of the digital transformation era? We are in it, yes, but halfway?
On line 69 of the page, the references to the CAA conferences are not well described in the bibliography.
Page 6, we come to the research question: "to what extent can geometric comparison methods in interaction with archaeological domain-specific information allow conclusions about ancient manufacturing techniques and places of manufacture?". It is a bit broader than your title.
The term digital twin is used, but it seems to me that we are dealing here more with digital models or digital shadows. But this is a detail.
It would be useful to detail the references to CIDOC CRM. There have been many recent publications, and it should also be specified which version of CIDOC CRM is used.
The methodological explanations are interesting, as are the results. I invite you to read the following paper, which is quite similar.
Poux, F., Neuville, R., Van Wersch, L., Nys, G.-A., & Billen, R. (30 September 2017). 3D Point Clouds in Archaeology: Advances in Acquisition, Processing and Knowledge Integration Applied to Quasi-Planar Objects. Geosciences, 7 (4), 96. doi:10.3390/geosciences7040096
In summary, it seems to me that the article would benefit from a refocusing on the main issue, less presenting things that are accepted in the community (especially in the introduction) which should allow for a shorter article presenting the innovative results (methodological and applicative in archaeology). This reduction is possible if the added value is emphasized. In its present form, it is more like a (very good) research report
Reviewer 2 Report
The paper deals with the up-to-date subject of involving machine learning methods to describe and analyze archaeological objects. It is undoubtedly innovative, of significant interest for the development of archaeological research methods, and can be published with minor corrections:
- line 178 - "inline", instead of "online".
- the term "ceramic genre" (line 209) is better to be replaced by "style"; the term "make group" (lines 210-211) is better to be replaced by "fabric group", because they are more often used to characterize archaeological ceramics.
- figure 15 - in the line "Application of logical rules" there is a mistake in the word "logical"
- there are doubts about the necessity of the publication of figures 22, 29 and 32 in the paper.
Reviewer 3 Report
This paper meets high scientific quality standards in many ways (methodological approach, references, data provided, etc.). In addition, it is properly written and applies a novel methodology in what regards the study of RSW appliqués. The research successfully conducts a semi-automatic comparison procedure of certain characteristics of the motifs represented in the appliqués, allowing the authors to determine their degree of geometric and visual similarity and to verify the existence of both local and global differences that may go unnoticed by means of traditional archaeological analysis. The final goals of the paper, as the authors point out, is to provide a new analytical procedure that allows archaeologists to be aware of some aspects of the appliqués that have been largely ignored up to now. By applying such a new AI procedure, archaeologists will be able to raise new questions in their investigation of RSW. Another interesting point in this paper is the effort made by the researchers to democratize the data and digital objects generated by making them available to the scientific community as a whole.
I totally agree with the authors in the fact that their work represents an optimal basis for the analysis of physical materials from both a geometric and visual point of view. However, the authors' work is directed to verifying that comparisons can be made with unprecedented precision by means of digital models of the motifs of the appliqués. The question that the authors must address here is “why is this information useful in the generation of knowledge regarding ancient societies?” There is hardly any reflection in the paper regarding the archaeological application of this methodology when it comes to infer aspects related to the origin or technology of archaeological ceramics. The authors vaguely state these significant concerns -which are quite complex in archaeology- in the manuscript and simply state that they “compare appliqués with the hypothesis that geometrically similar objects come from the same manufacturing place, thus enabling conclusions about the geographical origin and the manufacturing process (line 156)”. Therefore, the relationship between the geometric and visual similarity of the objects and their interpretation in terms of provenance and manufacturing process should be further developed in the text.
In this sense, the archaeological value of the methodology implemented can be significantly enhanced by stating more explicitly how AI would allow archaeologists to provide new information on past societies. In other words, there is a poor connection in the paper between “the geometric content with specialist archaeological content”. There is certain asymmetry in the paper in what regards the dialogue between computing science and archaeology in the definition of Archaeology 4.0. This unbalanced position can be clearly observed in the references included in the paper, in which there are hardly works on archaeological ceramics. If the authors are aware of this biased viewpoint of the paper but they still want to publish it, I suggest that it would be convenient to include at least this statement in the text, pointing out the need for further theoretical and archaeological reflection on the results provided.
I have also some ethical and moral concerns regarding the archaeological materials considered in this research. The ceramics studied presumably come from north Africa (Tunisia) and were purchased by a private collector (art dealer) (Line 279). In this sense, the authors must be aware that they were probably looted according to colonial archaeological practices. It is advisable that the authors provide further information regarding the origin of the vessels and discuss if they are ethically “clean”, if not they must justify why looted ceramics where analysed.
Specific comments and suggestions:
- End of line 24: References are needed here.
- Lines 181-185: There is no need to state the diverse subsections in this paragraph.
- Line 186: In Section 2 first explain in detail the archaeological materials selected in the case study then explain the methods applied in their study (General concept and so on).
- Line 216: Insert a dot at the end of the sentence, after the reference.
- Line 262: Include references here regarding the use of the mould in the production of Terrasigillata, specifically Sigillata C4 plates.
- Line 275: These numbers are confusing, which is the n? Please, clarify.
- Line 284: This brief definition of Digital Twin must appear before in the text, the first time you mention this concept in line 131.
- Line 287: I would rather say “…a copy of certain physical properties…”.
- Lines 384-385: what does it means here archaeological interpretation? Please, state which is interpreted in this step.
- Line 405: Briefly address how the semantic model deals with the typical influences on archaeological finds like wear, material fatigue and incompleteness which may disturb the original geometry of the objects.
-Line 665: Figure number wrong? Figure 23a?
- Line 698: You suggest here a common origin of the manufacture for both motifs. There is no reliability in such a statement. The provenance of a vessel is a complex issue in archaeology that usually requires to carry out several archaeometrical analyses of the vessels (e.g. petrographic, chemical, isotopic analyses). Please, further develop the relationship between motif similarity and origin here.
-Line 700: The same problem arises here, how do you know that differences between the motif geometries were produced while applying the appliqué to the carrier object? Experimental test can be carried out to address this issue in future works. Please, clarify your statement.
-Line 755: Further develop why from an archaeological point of view, there exists a certain probability for this assessment. Also provide references.
- Line 813: Which conclusions? Please be more specific here.
- Line 824: Which inferences regarding production and vessel use? Please be much more specific here.
Figures:
The figures are mostly correct and support the aspects discussed in the paper. However, the organization of the figures in the manuscript is not adequate at all and prevents a proper reading of the paper. I strongly recommend to relocate all of them so that each figure appears in the paper only when they have already been named in the text, never before the related text and in no case at the beginning of the sections.
- The way the authors represent the Schematic Model of the Archaeological Evolution (using a pyramid) in Figure 1 seems to me very inaccurate, since it promotes a viewpoint in which the different types of approximations are presented in an evolutionary sense. This implies a biased position since it does not consider the contributions and problems associated with each one of the paradigms and the social context in which they have meaning. All eras (including the analogue era!) generate relevant knowledge within the society in which they are inserted. In this sense, I suggest also to rework the text from lines 25 to 29.
- I suggest to replace Figure 9 (which does not provide relevant information) by a new figure that represents the two different types of appliqués recorded.